# Vigilance Decrement and Enhancement Techniques: A Review

**DOI:** 10.3390/brainsci9080178

**Published:** 2019-07-26

**Authors:** Fares Al-Shargie, Usman Tariq, Hasan Mir, Hamad Alawar, Fabio Babiloni, Hasan Al-Nashash

**Affiliations:** 1Department of Electrical Engineering, Biosciences and Bioengineering Research Institute, American University of Sharjah, Sharjah 26666, UAE; 2Dubai Police Headquarters, Dubai 1493, UAE; 3Dept. Molecular Medicine, University of Rome Sapienza, 00185 Rome, Italy; 4College of Computer Science and Technology, Hangzhou Dianzi University, Hangzhou 310018, China

**Keywords:** vigilance, vigilance decrement, vigilance enhancement, conventional means of enhancement, unconventional means of enhancement

## Abstract

This paper presents the first comprehensive review on vigilance enhancement using both conventional and unconventional means, and further discusses the resulting contradictory findings. It highlights the key differences observed between the research findings and argues that variations of the experimental protocol could be a significant contributing factor towards such contradictory results. Furthermore, the paper reveals the effectiveness of unconventional means of enhancement in significant reduction of vigilance decrement compared to conventional means. Meanwhile, a discussion on the challenges of enhancement techniques is presented, with several suggested recommendations and alternative strategies to maintain an adequate level of vigilance for the task at hand. Additionally, this review provides evidence in support of the use of unconventional means of enhancement on vigilance studies, regardless of their practical challenges.

## 1. Introduction

Vigilance can be defined as the ability to sustain attention and remain alert to a particular stimulus over a prolonged period of time. A multitude of industrial, military, medical, and educational tasks require continuous vigilance with varying levels of cognitive workload. Applications that need sustained attention include security personnel [1], employees tasked with monitoring surveillance cameras or baggage screening experts [2], driving vehicles [3], diagnostic medical screening [4], real classroom settings [5], and industrial and air traffic control [6,7]. Such tasks necessitate a specific bandwidth of arousal for maintaining an acceptable level of cognitive efficiency. An increased rate of target stimuli in stressful multitasking operations can lead to excessive cognitive workload and reduced cognitive ability. Similarly, monotonous stimuli can also lead to vigilance decrement and drop in cognitive efficiency. Past researchers have specifically reported that maintaining one’s vigilance in a stressful climate requires hard mental work [8]. A study has shown that target detection performance decreases by 15% in 30 minutes during a monotonous task [9]. This reduction in cognitive efficiency results in increased reaction time, error rate, and may even have fatal consequences. For instance, security personnel monitoring cameras may miss incidents suspicious activity, and drivers may cause traffic accidents. 

Vigilance decrement has been studied using overload and underload theories [10]. The overload theory has claimed that vigilance decrement results from the depletion of cognitive resources. According to the mindlessness theory [11], increased time on task causes people to gradually lose their focus on a task due to failure of the attention system. In contrast, the underload theory claims that the lack of stimulation of vigilance tasks cause the attention to shift, which results in performance decrement. In this perspective, attention resource is withdrawn either persistently or inattentively due to the routine nature of the task. Many factors are shown to influence the timing and magnitude of vigilance decrement. These include signal duration, source complexity, and declarative memory usage in the task [12]. 

Nevertheless, several intervention/enhancement techniques have been examined as potential solutions for attenuating vigilance decrement. The enhancement techniques aim to improve a subsystem in ways other than repairing a broken item or alleviating a specific dysfunction. Hence, reducing vigilance decrement with the use of cognitive enhancement techniques is very important and can lead to better safety, active learning, and improvement in overall quality of life. The literature contains a large number of conventional/traditional and mundane means for vigilance enhancement, including mental training [13], meditation [14], yoga [15,16], sports [17], exercise [18], caffeine [19], nicotine [20], diet and herbal extracts [21], chewing gum [22], and odor/fragrance exposure [23,24]. Reports of other unconventional (Unconventional means of enhancement are methods involving deliberately created nootropic drugs, neural implants, brain–computer interface and other new senses.) and more contemporary means of vigilance enhancers have also been reported, such as pharmaceuticals [25]; video games [26,27]; transcranial alternating current stimulation (tACS) and transcranial direct current stimulation (tDCS) [28,29,30,31,32,33]; tactile and rhythmic haptics [34,35,36,37]; integrating new challenges into the primary task [38]; music [39,40]; and binaural auditory beats [41,42]. Resulting studies have reported contradictory findings regarding the efficacy of the aforementioned enhancement techniques, which have hampered the progress of further investigations. To address these ambiguities, a comprehensive review of vigilance enhancement is undertaken in this paper to highlight the most promising direction for future research. 

This paper is organized as follows. Section 1 introduces vigilance, vigilance decrement, and vigilance enhancement. In addition, here we underline the motivation of minimizing vigilance decrement. Section 2 establishes the methods of search, inclusion and exclusion criteria, and the variables of interest. Section 3 and Section 4 review and discuss the effectiveness of the existing unconventional and conventional enhancement techniques, and further highlight their controversial findings/results. Section 5 provides a comprehensive summary of enhancement techniques and highlights their effects on vigilance. Next, Section 6 and Section 7 elucidate the challenges encountered and recommend several suggestions to enhance alertness or reduce vigilance decrement. Lastly, the concluding remarks are presented in Section 8. 

## 2. Methods

This section outlines the methods of search, inclusion and exclusion criteria, and the variables of interest in this study.

### 2.1. Search Strategy 

This review was conducted using the principles of Preferred Reporting Items for Systemic Reviews and Analysis (PRISMA) [43]. The following databases were searched for study publications, namely PubMed, MEDLINE, Science Direct, Web of Knowledge, IEEE Xplore, Google Scholar, and PsycINFO. The used search terms were the single terms of sustained-attention enhancement and vigilance enhancement. This was then also combined with at least one of the following terms; Caffeine, Music, Odor, Fragrance, Chewing Gum, Transcranial Brain Stimulation (i.e., tACS and tDCS), Binaural Beat, Haptic Stimulation, Distraction, Tactile, and Challenging Integration. In addition to the searching databases, the reference list for all selected articles was examined to identify any additional articles that might have been overlooked during the primary search.

### 2.2. Inclusion and Exclusion Strategy 

Manuscripts in English, original articles, and experimental studies were taken into consideration in this review. Furthermore, studies that evaluated the efficacy of attention, including alertness, vigilance, cognitive performance, speed, and reaction time, were also included. In contrast, those involving animals and older people were excluded to rule out the possible influence of cognitive impairments.

### 2.3. Variables of Interest 

The main variables examined in each article were (i) the amount of doses or the time duration of stimulation with each type of enhancer, (ii) vigilance test, (iii) number of subjects participating in the study, (iv) summary of results compared before and after enhancement, (v) attributes of the effects on vigilance, and (vi) comments on the findings.

## 3. Vigilance Enhancement Based on Unconventional Techniques

Table 1 summarizes the results and effectiveness of different unconventional vigilance enhancement techniques. The summary focuses on the type of enhancement with its quantity (i.e., dosage, frequency, or stimulation time), vigilance test, the number of subjects who participated in the study, a summary of the results, attributes of the effects on vigilance, and comments on the findings. “Positive” attributes of the effects on vigilance indicated that enhancement technique significantly improved vigilance. In other words, the type of enhancer reduced vigilance decrement. In contrast, “No Difference” suggests that the enhancer did not have any significant effect on vigilance and “Negative” attributes that the enhancer produced a negative effect on vigilance (see Table 1). 

### 3.1. Pharmaceutical Drugs 

Drugs such as nootropic, methylphenidate, and modafinil can enhance cognitive abilities, including one’s attention, concentration, and vigilance. They improve the brain’s oxygen supply by stimulating nerve growth or altering the organ’s neurochemicals [25,66,67]. Nootropics, in particular, have been shown to increase the concentration, alertness, and memory potential, while also potentiating the cognitive function in healthy subjects [45,68,69]. Meanwhile, methylphenidate and modafinil are commonly used by students to improve their exam performance and show superiority over their classmates. Their consumption is also seen in military personnel who need to remain awake for long missions and academics keen to maintain their performance [70]. Even a small amount/dose of modafinil can result in significant effects upon one’s vigilance [71]. A 100 mg dose of modafinil decreases the self-rating of depression and anger by 10 pilots during 37 h of continuous wakefulness, while improving the ratings for vigor, alertness, and confidence. Another study used 200 mg dose of modafinil on six pilots exposed to two 40-h of wakefulness and tested with helicopter simulating flights [72]. The pilots evaluated themselves to have less problems with mood and alertness compared to placebo. In a subsequent similar study that sampled 18 helicopter pilots, each had completed 15 flights and other evaluations during which they received lower dosages of 100 mg of modafinil (three doses at 4-h). They consequently reported having maintained alertness, feelings of well-being, cognitive function, and situational awareness. Such assessment has been consistently better than placebo usage and without the side effects of aeromedical concerns [73]. Interestingly, the effectiveness of modafinil on vigilance has been studied and compared with caffeine in many studies [44]. The technique has shown that the performance and alertness of 50 healthy subjects who administered 200 mg and 400 mg doses of modafinil are significantly improved compared to those obtained with 600 mg of caffeine. Although pharmaceutical drugs are depicted to improve performance and enhance vigilance, it remains yet unknown whether these drugs promote useful learning in real-life situations. Similarly, nootropics may be dangerous and have negative side-effects, such as headache, diarrhea, insomnia, fatigue, tremors, and nausea [74]. Additionally, some of these drugs are not approved legally, and thus, their consumption may be considered cheating. More studies aiming to reduce the side effects of modafinil are needed before its use for cognitive enhancement is approved in many real-life situations. 

### 3.2. Video Game

Nowadays, Action Video Game (AVG) has become a routine activity among children and adolescents. Green and Bavelier [49] define it as games that have “fast motion and required vigilant monitoring of visual periphery and simultaneous tracking of multiple targets.” Researchers have found particularly positive effects of AVG on basic mental processes, such as perception, sustained attention, memory, cognitive skills, and decision-making [49,50,75,76,77,78,79,80]. Various studies have shown significant improvements in the cognitive abilities of gamers compared to non-gamers, which are reported according to the detection rate of targets/accuracy, response time, and false alarm. Furthermore, Trick et al. [81] have highlighted AVG’s role in improving the ability of children and adults alike. Their task in the study was to keep track of a set of moving objects that were visually identical to other moving objects in the visual field. Similarly, a study [52] conducted on 179 subjects also demonstrated that the cognitive abilities of those who received computer game training are significantly improved. Such skills include sustained attention and preservation of multitasking tendencies. Besides, another work [76] has also found that AVG improves its subjects’ performance in locating targets that are rapidly presented and in tracking moving items in the presence of distractors. 

Furthermore, training, obtaining the knowledge of results (KR), and the expertise level of a gamer have been reported to contribute positively towards sustained attention [52,82,83,84,85,86,87,88]. Besides, receiving feedback regarding performance has also been found helpful. It not only carries an informational value but also offers motivational properties capable of influencing learning processes [83]. In particular, complete feedback (specifically Hit and False Alarm (FA) rates), provides the player with the information on signal characteristics in the task. When players are alerted to signals in this way they display the tendency to maintain sustained attention. Also, the proportion of correct detections will be high while the number of FA committed remains low. Other works have demonstrated that KR provided during a vigilance task results in improved performance over time, and it is transferred to a task similar to the training task [84,85,86]. In fact, KR obtainment has improved the overall performance during training, as well as during the subsequent test session in which feedback has been withdrawn [89,90,91,92]. Moreover, an analysis of previous studies has revealed that action video game players (AVGPs) show significant advantages over non-video game players (NAVGPs) in the context of vigilance performance, such as the proportion of correct detections, response time, false alarms, and more. Further workload analysis also indicates that AVGPs rate the task lower than NVGPs when it comes to total or global perceived workload [53]. 

Interestingly, it can be noted that AVG studies on sustained attention until recently have yet to find systematic gender differences [93,94,95]. For instance, Levy [94] did not encounter any difference in vigilance performance shown by male and female children aged between three and seven years. Similarly, Narra et al. [93] did not find gender differences in auditory and visual vigilance tasks for both child and adult participants. Meanwhile, Feng et al. [96] has stated that a 10-h training in AVG is sufficient to compensate for baseline gender differences in spatial attention and reduce the gap in mental rotation skills. Whether the primary difference is innate in nature or a by-product of lesser exposure to such kind of activities in women remains a matter of debate [50]. Additionally, despite AVG demonstrating its potential in enhancing cognitive abilities, the authors believe that the use of computer games as a means to enhance vigilance may not be very practical as it may take the attention off the current task at hand.

### 3.3. Transcranial Direct Current Stimulation

Transcranial direct current stimulation (tDCS) is a neuromodulatory technique that applies a small current in the range of 1 to 2 mA to the scalp, whereby it is considered safe for periods up to 30 minutes [97]. The tDCS causes changes in neuronal excitability via membrane polarization and alterations of the synaptic strength [98,99]. The direction of the cortical effects is mainly dependent on the polarity and waveform of the applied current. For example, anodal stimulation typically depolarizes the resting membrane potential and brings the neurons closer to their firing threshold. On the other hand, cathodal stimulation decreases neuronal excitability [100]. Therefore, changes in cortical excitability have lasted over one hour in duration and are demonstrated after a few minutes of stimulation [101]. Interestingly, an increasing amount of research can be seen in the recent past to emphasize the use of tDCS on cognition and clinical studies [31,102,103]. 

Recent studies have examined the effects of tDCS on vigilance and attention, as well as for alleviating problems at the workplace [32,58,102,103,104,105,106,107,108]. However, its impacts are dependent on the period of stimulation and electrode placement on the brain areas. For example, Nelson et al. [32] applied 1mA to the dorsolateral PFC (i.e., in the F3 and F4 positions) for ten minutes and reported vigilance enhancement in a simulated air traffic controller task. The tDCS has increased the target detection performance and operator discriminability, as well as increased the cerebral blood flow velocity and oxygenation when compared to a sham condition. Similarly, McIntire et al. [33] applied 2 mA to the left dorsolateral PFC (i.e., in the F3 position) for 30 minutes and used the right upper arm as reference while respondents undertook the Mackworth Clock Test and Psychomotor Vigilance Task. The results have shown improvements on vigilance to the same or a greater extent compared to caffeine. Meanwhile, Axelrod et al. [109] studied the effects of tDCS towards sustained attention task and reported an increment in mind wandering, but no effects were found regarding accuracy or reaction time. These findings indicate that tDCS may be well-suited to enhance performance degradation, highlighting the frontal lobe’s major role in mind-wandering behavior. Likewise, Cohen et al. [110] has applied tDCS to the parietal lobes of healthy adults during visual vigilant task with numerical characters and consequently showed an enhanced numerical proficiency. Another study that stimulated the parietal lobes during the training of a vigilance numerosity task has revealed improved and discernible effects up to one week [105,106,107]. The technique has been reported to have positive impact on improving the cognitive function of individuals with cognitive impairments. This is seen by the reduction of attention deficit symptoms [58,103,108] and alleviated vigilance problems at work [104]. 

However, not all studies have shown the positive effects of tDCS on sustained attention. Li et al. [111] applied 2 mA to the parietal cortex for 30 minutes and encountered negative performance due to tDCS during the final block of a reaction time task. Similarly, studies that employed the go/no-go task reveal no effects towards performance when increased demand is placed on the inhibitory control and shifting abilities are set [54,112,113,114]. Instead, they have reported the genetic factors that modulate the effects of tDCS on cognitive performance. Therefore, genetic variability should be considered in the design and analysis of future tDCS studies. 

In addition, stimulation parameters such as duration, intensity, frequency, electrode-position, and control settings can also modulate the outcome of the tDCS effect. More importantly, inter- and intra-individual differences, including genetics, age, gender, physiological differences, and baseline task performances, all imply the importance of a certain neural state. This state may determine the modulating effect on stimulated individuals through its interaction with tDCS [115]. It is suggested that the baseline state of each individual is different and that one’s receptivity to the tDCS changes with his/her baseline performance. It is also important for studies to choose an applicable baseline on which to evaluate the effect of tDCS. In addition, the task difficulty is another contributor to the state-dependent nature of the effects of tDCS. In a cognitive control task, the impact of tDCS was observed in the easy and medium difficulty conditions, but not in the case of the most difficult ones [57]. Thus, this review implies that tDCS effect is interactive and state-dependent. The task difficulty and consistent individual differences modulate one’s responsiveness to tDCS, while researchers’ choices of independent behavioral baseline measures can also critically affect how the effect of tDCS is evaluated. These factors are likely the key contributors to the wide range of variability in tDCS effects between individuals, stimulation protocols, and between different studies in the literature. Future studies using tDCS, and possibly tACS, should take such a state-dependent condition in tDCS responsiveness into account.

While tDCS appears to be quite versatile and noninvasive in nature, the risk of triggering epileptic seizures, and the effects of long-term measurement remains unknown. Consequently, it is still doubtful whether tDCS will ever be a practical and useful enhancement method. 

### 3.4. Music

Music plays a major role in the self-regulation of emotion and cognitive abilities, namely by modulating arousal and mood [116,117]. Various studies have shown that it is capable of improving attention and concentration, which result in better performance at work [59,118]. One study [118], in particular, has attempted to improve vigilance performance using background music in industrial settings, reporting vigilance improvement through music listening. This is done by creating and administrating a music program designed to increase tempo at peak fatigue times. Similarly, Davies et al. [119] showed that background music had increased the percentage of target detection in a visual vigilance task. This is especially apparent when the conditions are more complicated, subsequently suggesting a specific modulation of the alertness state that is caused by listening to music. Furthermore, Scheufele et al. [120] has introduced background music into job training and indicated that such element aids trainees to focus and complete their job assignments faster.

Moreover, Corhan et al. [121] investigated the effects of different types of music (such as “rock” and “easy-listening” music) on visual vigilance. They have reported that the vigilance performance is at the best state when the background stimulation is discontinuous and contains elements of uncertainty, such as in the case of rock music. The report on vigilance enhancement when using rock music led to another question. Can the improvements be attributable to the subjects’ familiarity with the types of music, rather than to their properties of discontinuity, rigor, and uncertainty? Researchers [122] have suggested that familiarity probably plays a larger role in the resulting performance. This is further substantiated by various works [123,124,125] that reported the act of listening to familiar music significantly increased arousal, and the detection rate on a vigilance task, as well as reduced vigilance decrement. The importance of the psychological characteristics of noise is also highlighted in determining the level of vigilance performance. One study [126] has suggested beforehand that such familiarity increases the arousal and improves vigilance performance regardless of the type of music being played. Another similar study on visual neglect has also underlined preferred music’s role towards enhanced patient performance during a perceptual report test, which is achieved by improving patients’ attention and vigilance [127]. These patients have shown enhanced visual awareness when completing the task accompanied by preferred music compared to non-preferred music and silence. This is suggestive of music’s influence in decreasing visual neglect by increasing the attention resource. Recent studies [60,62,63,128] have reported that music preferences pose a differential effect on miss rates. It has been found that those who like the music have a lesser tendency to show an increased false detection rate, while those who at least moderately like the music improve their detection rate. This results in the assumption that non-preferred music is distracting and the participants’ unawareness of such distraction leads to poorer performance.

Researchers who investigated the effects of specific types of volume and music on drivers’ vigilance have found that loud volumes affect simple vigilance, reaction time, and movements. Meanwhile, hard rock music impacts tasks involving concentration and attention [129,130,131]. Similarly, music tempo may also influence the listeners’ heartbeats and accelerate their performance. According to a recent survey [132], fast-tempo background music has increased one’s performance compared to listening to slow music when performing a vigilance task. Another example by Moris et al. [133] is indicative of playing music’s role in increasing the speed and accuracy of task performance by surgeons. Music has been revealed to reduce the heart rate, blood pressure, and muscle effort of operating surgeons while increasing the accuracy of surgical tasks concomitantly. This may be due to Mozart’s music that has displayed the capability to activate the cortical neuronal circuits related to attentive and cognitive functions [134,135]. In fact, different types of music (i.e., preferred, fast tempo, classical, etc.) can improve one’s performance in areas beyond vigilance tasks. Music enhances an individual’s concentration and cognitive function while it also allows them to maintain alertness. This results in improved detection rates and reduced false alarms [136,137,138].

Although music has been shown to improve performance during vigilance tasks, it can also degrade performance in other tasks [138]. Study in a cognitive test indicated that participants performed best when in silence, while in the background music is the second-best scenario for performance [139]. In contrast, having background noise yield the lowest results. When compared to silence, both background music and background noise negatively affect work performance. Additionally, a study in [61], revealed that playing background music with lyrics is likely to pose significant negative effects on the concentration and attention of a worker. 

One potential approach used for explaining the impact of background music on reading performance is the effect of lateralization. It is assumed that an increase in the activation of one’s brain hemisphere decreases the activation of the other region [140,141]. If background music activates the right hemisphere, the performance in tasks that need a highly activated left hemisphere, such as verbal tasks, could deteriorate. In this context, listening to music is considered dual-task processing. Thus, it not only involves listening but also the verbal abilities, and interferes with reaction time. Another candidate explanation for the negative impact of background music on memory processes might start with deliberations on the role of attentional limitations: listening to music while performing some cognitive task might distract attention from that task and therefore impairs performance, especially in tasks that require conscious efforts [142]. 

Previous studies have already observed that individual differences in personality and temperamental dimensions may play an important role in music preferences, exposure to different genres, music listening habits and use. However, even considering the strong emotional impact of music on humans, these affective responses are highly specific to cultural and personal preferences. Large individual differences are observed across individuals and on how music is experienced. 

Researchers must take into account individual differences when investigating the effects of music on employees’ fatigue and work tasks. The individual differences were essential in a pretest/post-test control group study carried out on 33 air traffic controllers [143]. The subjects completed trait anxiety and extroversion measurements, as well as a diagnostic stress inventory before the formal study. Results showed a significant reduction in stress level, when the subjects listened to music. Nevertheless, an interaction effect was exhibited for individuals with high trait anxiety and introversion, who did not demonstrate a reduction in anxiety. One of the possibilities is that the more an individual listens to music, the more likely they will experience an increasingly strong emotional response to music. In other words, the more knowledge one has of music, the more the emotional responsiveness. Without accounting for individual differences, the researchers may be missing an important work environment–person factor. Therefore, future studies regarding effect of music on vigilance should take into consideration the impact of valence, tempo, familiarity, and personal preferences during the design and analysis. 

### 3.5. Binaural Auditory Beats

A binaural beat is an auditory illusion perceived by the brain when two slightly different frequencies of sound are played separately into each ear. For example, playing a 250 Hz tone in the left ear and a 260 Hz tone in the right ear yields a beat with a frequency of 10 Hz. The brain then incorporates the process of the frequency following response (FFR) to follow along at the new frequency (i.e., 10 Hz). Low-frequency binaural beats are often associated with mental relaxation, while high frequency beats with alertness and attentional concentration [144]. Several reports have suggested that listening to binaural beats can reduce vigilance decrements [41,42,145]. 

A report has stated that exposure to binaural beats for 30 minutes leads to reducing vigilance decrement [41]. The study has tested participant’s performance during a 1-back vigilance test while listening to either delta 1.5 Hz, theta 4 Hz, or beta 16 Hz range binaural beats. They subsequently show improved in target detection and decreased false alarms, task-related confusion, and fatigue while listening to the beta binaural beat. Then, another study [42] investigated the effects of the beta binaural beat at 18.5 Hz on the vigilance task over a short period of five minutes. The authors of that study found an increment in the magnitude of beta electroencephalography (EEG) rhythm by 21%. Such increase of EEG rhythm ranges between 13 Hz to 21 Hz, with the highest increase in amplitude located at 18.5 Hz. Recently, another investigation [146] on three different binaural beats (i.e., 5 Hz, 10 Hz, and 15 Hz) and their relation with the vigilance task of delayed match-to-sample for five minutes was conducted. The results obtained have shown that the 15 Hz binaural beat increases the response accuracy and improves inter-region brain connectivity. Meanwhile, the largest EEG steady-state responses across the gamma band have occurred with a binaural beat of 40 Hz and primarily activated the frontal and parietal lobes [147]. Also, an investigation [148] on high-frequency gamma binaural beats at 40 Hz and vigilance performance has been done where the subjects undertake word list recall task for 30 minutes. Consequently, it indicates that listening to 40 Hz binaural beat for 20 minutes has enhanced the working memory function and improved the mood state. Similarly, another study [145] has recently shown that binaural auditory beats affect an individual’s control and visual attention. The participants listened to binaural beats at 10 Hz and 40 Hz while performing an attentional blink (AB) task, which generally assesses the efficiency of attention allocation over time. The AB has been reduced by the binaural beats for some participants, suggesting the specific impact of beats towards individual attention allocation over time. 

Conversely, the same conclusion as Lane’s [41] has not been achieved when using a different protocol. Goodin et al. [64], assessed theta at 7 Hz and beta at 16 Hz in a short 13 minutes period that presented in 2-minute tone/bursts. The study reported no significant differences in cortical frequency power during the period of binaural beats stimulation compared to using a white noise signal. Such discrepancies may be due to the use of short-time stimulation, while the choice of carrier tone may also influence the efficacy of beat stimulation. Besides, the robust effect of binaural beat towards vigilance may be obtained by the use of lower carrier tone and beat frequencies. Lorenza et al. [149] suggested that listening to the high-frequency binaural beat of 40 Hz results in bias with individual attentional processing, that is, towards reduced attention. The inconsistency compared to the results of previous studies may be attributable to several variables. They include the duration of the experiment, individual differences, carrier frequency, the specific frequency that fluctuates the cortical activities, cognitive test/vigilance task, attention to stimuli, sensory modality, spatial limitation (e.g., EEG), age, circadian rhythm, and hearing ability. Therefore, these factors should be taken into consideration when conducting an experiment, as well as during the analysis phase. 

### 3.6. Haptic Stimulation 

Haptic stimulation is comprised of two types of feedback, namely tactile and kinesthetic. The tactile feedback addresses the tactile perception from the skin (e.g., vibrations), whereas kinesthetic feedback is based on individual muscular efforts. Haptic stimulation has been established as a technique to enhance attention, whereby related technologies deployed in cars have revealed significant improvements for users’ driving performance. The stimulus is delivered at various locations in the car, such as the steering wheel, seat, or pedal [150]. These types of stimulations are thus used as an alarm to warn drivers of impending dangers such as veering off the main road and crossing the traffic light so that they are alerted to pay more attention. Various areas of driver’s body are also in constant contact with parts of the car (e.g., hands on the steering wheel), making those obvious locations for haptic stimulation during driving. In this context, discontinuous haptic stimulation acts as a positive distraction to the drivers. For example, when drivers start to lose their attention, tactile stimulation would reallocate their attention to the primary task (driving). Haptic feedback could effectively augment visual and audio feedback and tend to be quickly perceived by drivers. Interestingly, tactile feedback did not interfere with, nor was its effectiveness affected by, the performance of concurrent visual tasks. 

Several studies have investigated the effects of haptic stimulation on sustained attention. For example, a study in [36] stated that 15 Hz haptic stimulation has increased human sensorimotor rhythm band power and enhanced the short-term attention to vigilance task. Its results have shown that stimulation improves the level of attention to stimuli, reaction speed, and accuracy. Meanwhile, another study has been demonstrated that haptic stimulation is capable of enhancing the vigilance level of pilots during a stimulation flying task [65]. Its outcomes have also demonstrated that tactile feedback provides a higher detection rate and shorter responses times to unexpected events compared to visual feedback. However, continuous tactile feedback may disturb the drivers’ performance and increase stress levels. Another possible moderator is the habituation. Repetitive tactile stimulation may cause discomfort, increase workload, and shift the attention out of the task. Nevertheless, haptic stimulation is affected by age and may have a different impact on different genders [151]. It is a well-known fact that human haptic performance decreases as a function of age; however, the neural mechanism underlying these changes have yet to be elucidated. Therefore, future studies on vigilance should take several factors into consideration when developing the experimental protocol, design, and analysis. They include age, gender, time-on-task, and skin sensitivity, which can also be affected by various skin diseases. 

### 3.7. Cognitive Workload Modulation 

Cognitive workload modulation technique has been recently proposed to reduce vigilance decrement in video surveillance paradigm [38]. The technique is designed based on the integration of primary vigilance task with a visual noise, which is in the form of artificially simulated rain. The effects of incorporating challenging events (i.e., simulated rain) into the primary task have demonstrated a significant enhancement on vigilance performance and reduced reaction time to stimuli. Therefore, this type of enhancement seemingly increases the workload to a certain level, which in turn enhances the engagement of attentional resource. The culmination of such process results in improved vigilance level. However, the effects after stimulation on performance are yet to be explored, whereby more studies having a large sample size are needed to further support the technique’s effectiveness for reducing vigilance decrement. 

## 4. Vigilance Enhancement Based on Conventional Techniques

The following subsections discuss the findings of traditional/conventional enhancement techniques. We have included this section for a fair comparison with unconventional means in terms of contradictory findings, effectiveness, and cost. 

### 4.1. Caffeine

Caffeine is one of the most consumed psychoactive substances in the world [152]. The use of caffeine to stay awake and alert is a long-standing habit, with a daily amount of approximately 1.6 billion cups recorded globally [153]. Its stimulating effects on the central nervous system have been known for centuries [154]. Several studies reported that low to moderate doses of caffeine have positive effects on vigilance performance [66,155,156,157,158,159]. Its influence to affect the speed of reaction to task response has also been documented [158,160,161,162]. However, the duration of action it offers towards vigilance may depend on the amount of caffeine one consumes. Doses ranging from 32 to 256 mg have been shown to improve auditory and visual vigilance and detection rates, while they reduce the reaction time to stimuli in individual adults who performed vigilance task for several hours [158,163]. Furthermore, Fine et al. [162] confirmed and extended the aforementioned findings by demonstrating caffeine’s effect towards visual vigilance in 24 rested young males. It showed that consuming 200 mg of caffeine alone can significantly and consistently improve the number of correct responses while decreasing the reaction time to stimuli compared to placebo over two hours of testing. Meanwhile, Lanini et al. [164] reported that personalized individual caffeine dosages from 25 mg to 300 mg improved the vigilance of 60 young adults who performed psychomotor vigilance task. In this study, caffeine has significantly reduced the mean reaction time and led to less variable response times from the beginning until the end of a 10-minute performance of vigilance task. Similarly, Revner et al. [165] has indicated that 200 mg dose of caffeine can significantly improve the lane-tracking performance of subjects during a monotonous, two-hour afternoon, and early-morning drives in an automobile driving simulator. Moreover, Kilpeläinen et al. [166] reported on the effect of caffeine towards the vigilance and cognitive performance of 15 military pilots. These subjects have either received placebo or 200 mg of caffeine twice a day during an extended wakefulness of 37-hour period. The results consequently show that the caffeine group hit more targets compared to the placebo group, which is suggestive of the overall outcome of slightly better vigilance for the caffeine group. A recent study [160] tested the effectiveness of repeated four 200 mg dosages of caffeine on cognitive function and live-fire marksmanship in 20 soldiers whom performed psychomotor vigilance task (PVT), field vigilance, and logic reasoning tests during three successive nights of sustained wakefulness. The results have shown that caffeine allowed them to maintain their speed on the PVT, improved the detection of events, increased the number of correct responses to stimuli, and enhanced the response speed during the logic-reasoning test. Meanwhile, Doan et al. [167] investigated whether moderate doses of caffeinated tube food (200 mg) can enhance the performance of 12 pilots in a simulated U-2 mission. The study subsequently yielded outcomes indicating improved cognitive performance, vigilance and mood for pilots over a nine-hour duration. Likewise, McLellan et al. [168] examined the effects of 100 mg and 200 mg of caffeine on physical, vigilance and marksmanship tasks for 30 soldiers during a sustained 55-hour field exercise. Their conclusion was that caffeine is an effective strategy to sustain vigilance and psychomotor capabilities during military operations. Besides, similar improvements have been found in 86 hemodialysis patients who undertook habitual coffee consumption when they received 120 mg dose of caffeine [169]. Hemodialysis patients frequently have cognitive impairment with marked executive dysfunction and reduced attention due to defective processing of relevant perceptual stimulus. The results have validated its beneficial impacts on the cognitive function of hemodialysis patients, due to its direct enhancement of attention, concentration, and vigilance. The positive effect of caffeine under states of reduced alertness is proven to be quite consistent and the ability of caffeine to sustain attention may be important for patients undergoing dialysis treatment.

However, the effects of caffeine towards higher cognitive functions remain debatable. While it may increase a person’s ability to stay awake, it does not necessarily help in making good decision. This is ultimately the skill that is critically important in military and driving occupations. Furthermore, it is unclear whether caffeine has positive or negative effects on mood and emotional state. Thus, it calls for further investigation into another form of fatigue countermeasure to enhance alertness and performance. Despite the fact that some studies show the effectiveness and safety of caffeine consumption, vulnerable populations may be harmed by excessive intake as it may increase the risk of dehydration, abnormal cardiovascular function, headache, sleep disturbance, and substance use [170,171,172,173,174]. In addition, the research findings for caffeine on vigilance are still conflicting, whereby the effects are dependent upon individual and amount of doses consumed. Frewer et al. [156] reported that 500 mg dose of caffeine impairs the performance of 12 healthy subjects at 45-minute post-administration time. Nevertheless, low doses of 40 mg to moderate 300 mg doses may seemingly improve alertness and vigilance, as well as reduce reaction time. However, less consistent effects are observed for memory and higher-order executive function, such as judgment and decision-making [175,176,177]. 

Individual differences in response to caffeine are also commonly reported [178,179,180]. Researchers often point to substantial individual differences in behavioral responses to explain inconsistencies among experimental findings. These individual differences are often attributed to differences in subjects’ regular use and the number of doses. It has been thought that consumers of high daily amounts of caffeine might have acquired tolerance to the behavioral effects of caffeine, and are therefore less sensitive to its effect on behavior. However, little is known about the relationship between subjects’ regular caffeine intake and their behavior sensitivity to an acute dose of caffeine. There is some evidence that daily administration of 300 mg of caffeine can produce tolerance to its subjective effects in humans [181]. 

There is no evidence to date that demonstrates that normal consumption levels of caffeine relate to acute behavioral responses in human. Fillmore et al. [182] examined subjects’ reports of typical daily consumption in the study of individual differences in response to caffeine. The study concluded that subjects’ caffeine use could not account for individual differences, but subjects’ expectancies about the effect of caffeine did predict individual differences in response to the drug.

### 4.2. Fragrance Administration

The sense of smell plays an important role in the physiological effects of mood, attention and working capacity. Multiple studies have reported that introducing certain types of fragrance can improve attention and vigilance performance, while reducing stress [183,184,185,186,187,188,189]. The administration of high-valence odors like peppermint and cinnamon has particularly shown improvements in vigilance and reaction time to stimuli [190,191]. Furthermore, looking into the effects of olfaction (Olfaction is a chemoreception that forms the sense of smell) on human behavior while doing cognitive vigilance tasks has resulted in various works. A study [192] investigated the effects of ambient scents that are administered while doing vigilance task of anagram and word completion tests. The study demonstrated significant reduction in vigilance decrement, as well as improved attention, mood and cognition. Another study [193] has attempted to explore the effects of olfaction on human behavior while doing three different vigilance tasks. These tasks include clerical tests typing, memorization, and alphabetization, which are undertaken in either a nonscented or peppermint-scented condition. The results show significant improvement in the speed and accuracy for the typing and alphabetization tasks, while the peppermint scent is associated with increased performance. 

Furthermore, study in [183], found that performance and accuracy of detection in a vigilance task increase when the subjects receive short whiffs of either muguet or peppermint fragrances. This is compared to those who received only whiffs of pure air. The vigilance task has been particularly involved with the detection of two lines presented with a separation of 10 mm instead of a separation of 12 mm. Meanwhile, Jones et al. [194], upon examining the effects of pleasant and unpleasant fragrances alike with the same vigilance task used by Warm et al. [183], have indicated that an exposure to pleasant odors enhance vigilance performance. In line with such conclusion, further exposure to pleasant ambient fragrance (i.e., lemon aroma) by people who are performing simulating driving task [185] revealed significant enhancement for driving performance and increased level of alertness. Similarly, Matsubara et al. [24] has investigated the effect of volatiles emitted from the leaves of *Laurus nobilis* at low and high doses towards vigilance performance in a visual discrimination task; they found that low concentration of the leaves has resulted in high levels of vigilance performance and improved detection rate and physiological arousal.

However, the alertness-related properties of peppermint have been depicted as unreliable. For example, Moss et al. [195] reported a significantly higher alertness when people are exposed to peppermint compared to ylang-ylang. Regardless, it is not significantly greater compared to the non-odor condition, which suggests that peppermint does not increase alertness by itself. Meanwhile, other studies [35,196,197] reported no effect by peppermint in detecting the visual signal in a vigilance task. A possible explanation for such conflict may be attributed to habituation. In case of [183], the odor has been presented in 30-second bursts every five minutes via a mask, whereas Gould et al. [196] has opted for the odor to be present throughout the 20-minute vigilance task as performed by Moss et al. [195]. Moreover, other factors regarding the effects of fragrance on vigilance may be linked with the concentration, mood state, time-on-task, and experimental protocol. 

One possible reason for the lack of consistency in previous findings is that existing individual differences in olfactory sensitivity have been overlooked. In particular, sensory sensitivity plays a major role in shaping the response of such individuals. A few studies have explored individual differences in olfactory sensitivity and found approximately 20 per cent of the population self-reported as having a heightened sensitivity to scent, in comparison to 70% who self-categorized their sense of smell as normal [198,199]. Recent findings revealed an automatic suppression mechanism for individuals sensitive to smell [200]. Combined, these findings suggest that a significant proportion of the population is sensitive to smell. Another study [201] used the impact of odor (AIO) scale to measure the importance of smell on an individual’s liking of people, places and products. It was found that high AIO scores were associated with odor-related memory, and attention to odor. This also influenced their liking or disliking of people based on their odors. Similar Lin et al. [202] investigated these effects across individuals based on their olfactory sensitivity. It found that odors play a significantly stronger role in individuals sensitive to smell on perceptions of place and people in a service setting, as well as on cognitive processes such as attention, memory and emotions.

### 4.3. Chewing Gum 

Chewing gum is traditionally used to prevent sleepiness during work, learning, and driving, which in turn suggests a link between chewing and vigilance enhancement. Various studies have reported the positive effects of chewing gum on vigilance performance, alertness, and stress [22,203,204,205,206,207,208,209]. The action maintains and increases one’s self-rated level of alertness, which has been established as a contributing factor to enhanced vigilance [205,210]. The investigation by Allen et al. [206] has been undertaken by looking into the effects and after-effects of chewing gum on vigilance, mood state, heart rate, and brain oscillation (measured by EEG). The results have shown that chewing gum reduces the reaction time to stimuli, increases the detection rate of hits, heightens the heart rate, and increases the EEG beta power at F7 and T3, immediately after chewing. Note that the mood state and time-on-task are similarly important factors in the psychodynamic of the action. Meanwhile, another study [211] has opted to examine the effect of chewing gum for thirty minutes on healthy adults while performing a vigilance task. Their conclusion is that chewing gum decreases the reaction times to target stimuli during the first 20 minutes of vigilance task when compared to the control group that consisted of people who did not chew gum. Another study [208] has also investigated the effects of chewing gum for thirty minutes on subjects performing auditory vigilance task. The task involves listening to a random presentation of digits from one to nine, at the rate of one per second. The obtained results indicated a significant decrease in the reaction time and improved alertness in people who chewed gum. Likewise, another work [212] has utilized a variety of techniques to study the links between chewing gum versus performance, alertness, and stress. The results again suggested that chewing gum can reliably maintain one’s alertness, enhance their performance at work, and reduce vigilance decrement. It may be concluded that the improvement on sustained attention, alertness and the reduction in vigilance decrement in the context of chewing gum may be due to the gum-enhanced delivery of glucose to the brain. This is a direct result of insulin secretion during mastication [213]. Unlike other enhancement techniques, chewing gum is an easy method for modifying cognitive function on a daily basis and does not demand any physical or mental efforts. 

However, the reported effects of chewing gum on sustained attention have been rather inconsistent and yet to be fully established. A recent review of 22 studies [22] has highlighted that 64% of studies indicated the positive attributes of effects on attention. In contrast, 5% showed negative attributes, 23% showed both positive and negative attributes, and the remaining 2% showed no significant effects. Such observations may be attributable to the variations of experimental protocols, brand of gum, familiarity with gum, and the method used for analysis. Additionally, the effect of chewing gum on attention seems to be influenced by time-on-task, i.e., the duration of action may not last 20 minutes [214] or 30 minutes [211]. 

## 5. Comprehensive Summary 

After reviewing the nature of the stimulating tasks and the contexts in which these studies were conducted, we now present a critical integrating summary on the effectiveness of enhancement techniques on vigilance. This section focuses mainly on highlighting the attributes of effects on vigilance (positive, negative, or no significant impact on vigilance). Highlighting the effects of various enhancement techniques on different types of vigilance tasks may help researchers/scientists to select the appropriate enhancement technique in the context of their study and may help in reducing the contradictory findings. The comprehensive summary is derived from all the studies discussed in the previous sections, including the ones listed in Table 1. We categorize the vigilance tasks into two types: monotonous and complex. We define the monotonous tasks as the ones that require sustained attention, such as visual monitoring and target detection. On the other hand, we define complex vigilance tasks as the ones that are more cognitively demanding, e.g. comprehension and working memory. We have conducted this review in the context of job/work, monitoring/ surveillance, driving, learning, cognition and memory tasks, typing, shopping, and sports. The percentage improvements in vigilance measured by reaction time (RT) for each enhancement technique are depicted in Figure 1 and Figure 2 for monotonous and complex vigilance tasks, respectively. Positive RT improvements indicate the enhancement technique reduced the reaction time to stimuli while negative RT improvements increased or impaired the reaction time. Zero RT improvements indicate that the enhancement technique did not have any significant effect on vigilance. For each enhancement method, the error bar represents the standard deviation from the percentage RT improvement mean of the extracted data from various studies.

To the best of our knowledge, when simple monotonous vigilance tasks were considered, all enhancement techniques showed a positive impact on vigilance. The techniques showed average improvement varying between +8% and +18% as measured by the percentage reaction time (RT) improvement depicted in Figure 1. The highest improvements (more than +15%) on monotonous vigilance tasks were reported when video game and transcranial direct current stimulations were used [32,33,49,52]. Haptic stimulation (HPS), modafinil, cognitive work-modulation (CWM), tACS, chewing gum (CG), and caffeine [28,29,35,38,46,164,206] showed an improvements of more than +%10. However, music, binaural beats (BBs), and fragrance produced improvements of equal or less than +10% in the simple monotonous vigilant tasks [41,123,124,125,193]. 

The large improvements in vigilance associated with action video games are consistent with prior work [215] that associate video game play with neuroplasticity in the afferent brain regions and neural networks. This involves specifically the areas associated with empathy, motor executive function, decision-making, and emotion regulation. Such modulation may thus enhance the communication between brain regions responsible for attention and results in improving performance. In addition, engaging action video games increase the arousal level of players by the adrenergic route, which in turns make the motor behavior more responsive decreasing the RT to the incoming stimuli and increasing attentional abilities [216]. In particular, research has suggested that playing AVGs can lead to improvements in perceptual [217], visuospatial [96], perceptuomotor [218], and attentional abilities [49], and that such improvements can also extend to cognitive control functions such as cognitive flexibility [219] and working memory updating [220], but not inhibitory control [221]. Similarly, the improvements in vigilance using tDCS may be due to altering the membrane potential and spontaneous firing rates [222]. This in turn increases the connectivity in the prefrontal cortex, dorsolateral and other remote brain regions. The modulation of targeted brain cortices led to improved performance. Likewise, the positive impact of HPS on vigilance is due to the changes induced on somatosensory cortex such as representational map reorganization [150]. HPS studies have demonstrated that the stimulation of tactile afferent fibers and essential neuroanatomical elements of affective touch activates specific brain areas. This activation pattern is influenced by subject’s attention. The changes in the functional brain network enhance the information transfer and attention. 

Additionally, the improvements on vigilance with modafinil is consistent with studies on adaptive inhibition, see review [223]. Modafinil exerts its action by competitively binding to the dopamine transporter as well as by inhibiting norepinephrine uptake. This produces an overall elevation of catecholamine levels and potentiation of adrenergic neurotransmission. Stimulants enhance attention by increasing neuronal activation or by releasing neuromodulators, facilitating the synaptic changes that underlie learning/attention. Overall, the increment in the neural activations and connectivity patterns improved performance on simple and complex vigilance tasks. 

Cognitive work-modulation increases the workload to a certain level, which leads to increasing the engagement of attentional resource and enhance performance on simple vigilance tasks. In the same way, the positive impact of tACS on vigilance can be explained by observing that tACS targets the brain’s natural electrical oscillations, which represent neuronal patterns of communication throughout regions of the brain [224]. This is unlike tDCS, which targets brain structures, such as particular regions of the cortex. Therefore, it permits physiological entrainment through frequency stimulation at nearly imperceptible current strengths. Consequently, the improvements of BBs on vigilance may be due to BBs producing phase synchronization across the cortex [147]. The increase in phase synchronization in the auditory cortex facilitates the neural communication, promotes neural plasticity, and enhances the overall performance. 

The positive impact of CG on vigilance could be explained as due to the fact that it increases the heart rate, which leads to increasing the flow of nutrients, such as glucose, to the brain [225]. The positive effect of CG on vigilance tasks may also be due to either (1) CG restoring arousal after a vigilance task reduces arousal to a suboptimal level or (2) CG reducing arousal after a vigilance task heightens arousal to an excessive level [206]. Correspondingly, the improvement of music on simple vigilance tasks is in line with a study that used music to regulate people’s mood and arousal level [226]. Literature has reported listening to preferred music reduces stress during driving and lowers emotional arousal under frustrating circumstances, such as heavy road congestion [126,143]. In this context, comfortable volume of background music exposure improves one’s performance while performing simple vigilance tasks. One may also note that music has a positive impact on emotional reactions and achievement in sports. Moreover, the positive impact of fragrance on vigilance could also be explained due to its effectiveness in alerting mood and feelings [183]. Relaxing fragrance affects performance efficiency and feelings, since subjects who are tense and uncomfortable may find it hard to concentrate on the task.

On the other hand, when complex tasks are considered, smaller improvements with contradictory findings on average of −2% to +7% are found as shown in Figure 2. For this type of vigilance enhancement tasks, Modafinil, tDCS, tACS, and VG showed the highest improvements to vigilance compared to the other enhancement techniques. Interestingly, tACS has shown consistent positive impact on the two types of vigilance tasks (simple and complex), with an average improvement of more than +6% on complex vigilance tasks. Similarly, BBs, HPS, CWM, and fragrance showed both positive impact and no significant impact on vigilance. Unfortunately, the number of studies for the majority of these contexts and tasks was small, rendering the respective conclusions somewhat unreliable. Likewise, CG and caffeine reported positive and negative impacts on vigilance with average improvements of more than +3% on complex vigilance tasks. Meanwhile, some of the complex tasks showed impaired performance with caffeine and CG [227,228]. This may reflect the effects of increased alertness or chewing interfering with subvocal rehearsal. 

Modafinil appears to enhance sustained attention, learning, and memory. Negative cognitive consequences of modafinil intake were also reported in a small minority of complex tasks, though not consistently on anyone [223]. Although there is impairment of the reaction time to stimuli in some tasks, low doses of modafinil have been shown to enhance working memory in healthy test subjects, especially at moderate task difficulties and for lower-performing subjects. Even though the mode of action of modafinil is not yet understood, it may be noted that modafinil enhances adaptive response inhibition, making the subjects evaluate a problem more thoroughly before responding, thereby improving the overall performance, reaction time, and accuracy. 

Negative consequences of VG were also reported in a small minority of complex vigilance tasks especially with children at school [229,230]. Playing VG showed decreased students’ performance at school, which might be due to developing attention problems and poorer sleep quality. In this context, attention deficits and poor sleep could both plausibly impair academic performance. Moreover, music has been found to negatively impact complex vigilance tasks. For example, in comprehension it reduced performance down to −7%, regardless of the type and condition (fast tempo, classical, and familiarity) of the music [60,61]. Positive effects (better results with music) were found only for simple math tests [231]. Background music with complex vigilance tasks acts as a distraction when it comes to human vigilance performance. Even though music has been shown to benefit driving performance and behavior, it may still be a major distraction and detrimental to driving abilities, e.g., at loud volumes. It is suggested that loud volume might influence vigilance due to its greater processing demands on the central nervous system. Attention may be deterred from the primary task/task at hand, thus causing an impaired RT. An alternative reason is that such high sound volume may cause an anxiety effect within subjects. It is well documented that chronic exposure to noise increases stress levels. Another possible moderator is the habituation. Since music media are increasingly available everywhere, its effects may diminish over time. The review highlights that the negative impact of music on complex vigilance tasks (such as comprehension) is a general effect. 

Looking at the nature of various vigilance tasks, it is quite safe to make a conclusion regarding the effectiveness of enhancement techniques. Consistent positive effects were found only with simple vigilance tasks. VG, tDCS, tACS, and modafinil work well in enhancing vigilance regardless of the type of tasks compared to other enhancement techniques. However, there is variability across subjects in the extent to which stimulation modulates behavior. This provides a challenge for the development of applications. For example, a large part of variability in the after-effects of motor cortical tDCS may be due to the interindividual differences in the electric fields and size of electrodes. In this context, we anticipate that individualized electric field dosimetry could be used to control the neuroplasticity effects of tDCS. It is suggested that an effort should be made to develop more specific theories about the impact of cognitive enhancement and to increase the methodological quality of relevant studies. In fact, individual differences in response to the aforementioned vigilance enhancement techniques are also reported. These individual differences are often attributed to differences in subjects’ routine use, number of doses, culture, and preference. There may be other individual differences, for example, personality and age that are known to influence performance. Researchers have thoroughly investigated other variables, including temporary individual and stable variations. Temporary differences comprise coping strategies and fatigue, while stable differences include sleep behavior, gender, and introversion [70,232,233,234]. Various research groups have proposed alternative neurophysiological and biological biomarkers to investigate vigilance tasks. They include electrical neural activity, salivary melatonin levels, circadian rhythm, heart rate variability, and cerebral blood flow. Some of these neurobiological implications can be influenced by training, such as deep breathing and biofeedback exercises. Hence, it is possible that given such training, individuals may be able to handle more prolonged or more complex vigilance tasks; however, few such studies have been performed. 

## 6. Challenges

Conventional and mundane means of vigilance enhancement are often well-established techniques and are culturally accepted by societies. They include mental training, yoga, sports, exercise, meditation, nicotine, herbals, caffeine, fragrance/odor, and chewing gum intake. These elements have been investigated for decades and thus the chance for further improvements may be limited. Their intervention occurs across the wide cognitive domains of memory, perception, attention and understanding. Conversely, unconventional techniques are new and have the tendency to evoke social concerns due to their novelty and experimental-level phase they are currently in. Unlike conventional means, the interventions of unconventional techniques can be specific to a narrower domain, such as sustained-attention or vigilance. Although these techniques are new, they are expected to be integrated in the ordinary category of human tools alongside the development of technology and societal experiences. However, challenges are rife with regards to the use of conventional and unconventional means of enhancement alike. Different types of enhancements pose different social challenges. The following subsections discuss the challenges that cause major concerns when developing the enhancement techniques. These discussions are from different perspectives, such as safety, health, cost, sensitivity, response time, adhering to ethical approval, and translation to real-life practice. Table 2 summarizes and compares the effects of obstacles across different enhancement techniques. 

### 6.1. Safety 

The safety of conventional enhancement techniques has been reported in many studies [235]. Improving ones’ ability is of interest in several contexts, e.g., in learning, driving, shopping, sports, etc. However, enhancement may lead to inequality objections, which in turn have a negative impact on society. Psychological training and education are commonly considered to be safe. Note that their long-term use may affect some neural organizations. In fact, all types of conventional enhancements/interventions carry some risks, while their benefits tend to be subjective in nature. Therefore, individuals should choose their own preferences for adjustments according to the risks and benefits. Unconventional means are relatively new, and hence render the concerns about their potential use, safety, efficacy, and social consequences to require further exploration. These types of enhancers may face hindrances in terms of acceptable risk to test subjects as well as the reliability of research. For example, tDCS has been used for enhancement and treatment despite the limited knowledge regarding its safety so far. In this context, the safety of brain stimulation depends on the strength of current, duration of stimulation, and the size of electrodes [97]. Thus, long-term stimulation of tDCS may seemingly cause discomfort, tingling and itching in healthy individuals and patients [236]. Nevertheless, safety concerns surrounding the effects of long-term usage of conventional and unconventional means of vigilance enhancement indicate that the safety vs. efficacy needs further investigation. At the same time, it is very important to establish a baseline level of acceptable risk for interventions. This can be done in comparison to existing risks that are acceptable by the society, such as the risk of smoking. In addition, privacy and data protection are very important in the context of safety issues and should be taken into consideration when developing a cognitive enhancer. 

### 6.2. Health

Cognitive interventions/enhancements, in general, are safe in nature and characterized as noninvasive tools to modulate neural activity in humans. However, some interventions may cause adverse effects. A recent study has reported that nootropics such as modafinil induces the elevation of dopamine level and may lead to drug abuse and addiction [237]. Modafinil may also affect the autonomous nervous system by increasing the resting heart rate and blood pressure. This limits its usage to individuals without any heart disease. Nevertheless, classifying modafinil as an addictive substance remains controversial and its waking mechanisms are not fully elucidated yet. 

To date, all reported adverse events of tDCS have been transient rather than persistent in nature; no serious adverse effects have been reported in healthy subjects or patients [238]. On the contrary, excessive intakes/doses of caffeine is found to be correlated with an increased risk of dehydration, cardiovascular function, headache, intoxication, sleep disturbances, and substance abuse [44,153,170]. On the other hand, the neural responses to music affect the emotion regulation of humans, whereby those who process negative feelings with music react negatively to aggressive and sad music. This may lead to health issued if listened for prolonged durations [239]. Additionally, constant chewing gum action can lead to a jaw problem called temporomandibular disorder [240]. It is worth noting that binaural beats and haptic tactile sensations are relatively new in the context of cognitive enhancement compared to other approaches. Therefore, the effects of these interventions upon human health have not been fully investigated yet. 

### 6.3. Cost

The cost varies across different enhancement techniques and depends upon their effectiveness, reputation, safety, accessibility, and efficacy. The methods that are used to correct specific pathological element or defect in the cognitive subsystem such as genetic engineering are expensive. Meanwhile, interventions that improve a subsystem in some way other than remedying a specific function are less expensive. In particular, unconventional enhancement methods may potentially reduce the overall cost of enhancement compared to the conventional methods. Although unconventional methods are new and still in their experimental stage, their optimal cost cannot be easily identified as it is constantly reducing due to technological development. Table 2 compares the estimated cost of existing conventional and unconventional enhancement techniques in the scales ranging from low to moderate to high, with respect to a single dose.

### 6.4. Assessment Methods

Evaluating the effects of interventions is traditionally achieved using questionnaires or by interviewing the subjects. This type of assessment may be subjective in nature and may not reflect the real-time situation after the experiment. Thus, using an objective assessment method is very important to reflect the improvements in real-time situations, namely, by providing quantitative information. Note that advanced neuroimaging modalities, such as EEG and functional near-infrared spectroscopy (fNIRS), may be the best candidates for assessing the effect of interventions towards alertness and vigilance. These practical and objective techniques may help one to understand the underlying neural mechanism of vigilance. It may also aid the optimization of the types of enhancements/interventions for modulating the cognition and mood states. 

### 6.5. Translation to Real-Life Practice 

Translating enhancement techniques into real-life practice may depend on their degree of complexity, acceptance by societies, accessibility, license, and tolerance for integration with technology. Those that reflect the potential personal and social benefits will reach more people and be used in many applications. Therefore, more investment for the research and development of cognitive enhancement will improve its practicality and efficiency. Table 2 summarizes the extent of ease and difficulty for one to translate conventional and unconventional enhancement techniques into real-life practices. 

### 6.6. Response Time

The effectiveness of conventional and unconventional means of enhancement on vigilance may vary across techniques and subjects. People who received training on a particular intervention may perceive its effects faster than those who undergo interventions without any prior training. According to previous vigilance studies, the effective response time varies across techniques in the range of minutes as shown in Table 2. However, the after-effect duration of enhancement is yet to be fully explored for most of the enhancement techniques as seen in Table 2. The studies show that the action after-effects are the longest for modafinil, followed by tDCS, caffeine, and chewing gum (Table 2). It may be noted that the after-effects in tDCS depend on the stimulation time. For example, a stimulation of 9 min, 13 min, and 30 min can produce significant after-effects in neural excitability for 30 min, 90 min, and 6 hours, respectively [33].

### 6.7. Adhering to Ethical Approval 

All types of cognitive vigilance enhancement require a review and ethical approval from their respective institution or organization. There may be different ethical standards across the globe due to cultural, social, and political inclinations. Therefore, getting ethical approval for some interventions are difficult or time-consuming, e.g., for pharmaceutical enhancement techniques [235]. Although medical interventions are conducted on both healthy and patient subjects, the process of obtaining ethical approval in healthy subjects may be easier, whereas patients may need their relatives to sign the consent on their behalf. Table 2 summarizes the extent of ease or difficulty for ethical approval obtainment in most conventional and unconventional enhancement techniques. 

## 7. Recommendation

The following subsections present some recommendations for maintaining an adequate level of vigilance for monotonous as well as complex vigilance tasks. They also suggest some strategies to improve the experimental protocol and data analysis at the individual and group level of vigilance studies.

### 7.1. General Recommendation

Testing for developed cognitive enhancement techniques is usually conducted in a quiet and controlled environment, such as a laboratory. Researchers should test the developed enhancer/intervention technique in studies conducted in a more realistic scenario. The ultimate criteria of efficacy would be the various forms of real-life successes rather than narrow psychological laboratory tests. Note that portable and wearable technologies may allow unobtrusive monitoring of behavioral responses, diet, and drugs in a large sample of population. Here, data mining could also help one to determine the effects of enhancers on reducing vigilance decrements, given enough data. 

This paper recommends the use of unconventional enhancers due to their reliability, newness, and computerized integrity, as they are less ethically controversial at present. Furthermore, several suggestions have been highlighted for future works that may reduce vigilance decrement. First, a combination of different enhancement methods may perform better than any single method, especially in the workplace settings where a wide variety of tasks have to be performed. For example, a combination of rhythmic visual and or/auditory stimulation with haptics-based brainwave entrainments may complement each other and reduce vigilance decrement. Thus, future studies may implement the multisensory aspects of vision, auditory and touch in a prolonged monotonous task to reduce vigilance decrements. Second, it is advisable for vigilance to be assessed using portable and wearable advanced neuroimaging with good temporal and spatial resolutions. This will aid in understanding the neural mechanisms underlying vigilance decrement and after enhancement. 

### 7.2. Recommendations for Future Experimental Studies 

A number of aspects related to the experimental design should be considered by researchers when investigating the effects of interventions/enhancements on vigilance. The following section summarizes the elements that should be taken in consideration during the design and analysis phase of experimental vigilance studies. 

#### 7.2.1. Duration of Measurement and Enhancement Stimulations 

Some studies do not consider the duration of data acquisition. Short-time measurement may not induce vigilance decrement while longer acquisition time may induce fatigue and pain. Vigilance decrement typically requires performance over an extended period of time and enhancement techniques should demonstrate improvements over the entire length of experiment. Thus, researchers in future studies should consider the duration of experiment. At least 30 minutes of measurement should be undertaken to confirm vigilance decrements, as per the prior studies. 

#### 7.2.2. Resting Period during the Acquisition 

Future vigilance studies should not involve resting breaks during the acquisition. Resting break during a period of continuous monitoring is an effective countermeasure for restoring the performance in auditory and visual vigilance tasks [37]. Therefore, it is difficult to ascertain whether the improvements were due to the resting break or from the enhancers. A continuous measurement is henceforth suggested over short recordings in successive blocks with resting breaks in between. 

#### 7.2.3. Time of Day, Spatial Location, and Visual and Sound Level

Workload and vigilance are affected by stress hormones, which are highly produced in the early mornings and lesser in the afternoons. Future studies should take into consideration the effects of stress hormones and circadian rhythm to one’s vigilant state. In addition, spatial location for measurement and enhancement stimulations may play a major role. The brain regions that are highly affected by vigilance should be identified prior to enhancement stimulations for a better result and reduction in complexity and cost. Researchers should also consider the impact of vision and auditory abilities, especially in enhancement techniques involving sounds and other multisensory elements. 

#### 7.2.4. Vigilance Task Considerations

Easy tasks can cause habituation and result in no enhancer effect compared to normal control conditions. Therefore, the choice of vigilance task needs to be carefully considered. Those that involve planning, executive functions and decision-making are highly recommended as they reflect vigilance in real-life situations. Moreover, some studies do not attempt to relate neural activation to task performance, thus suggestive that future studies should put in an effort to link the two elements in their analysis. 

#### 7.2.5. Pre- and Post-Test Measurements

It is very important for future works to include pre- and post-test measurement analysis for activation changes and comparison between conditions before, during and after and intervention. It is very difficult for one to ascertain the influences of vigilance task and enhancers alike, without a baseline. Therefore, the postmeasurement may help in highlighting the effectiveness of the developed enhancer with respect to the duration of action. It is also advisable for researchers to report the manner in which the baseline recordings are taken for evaluating emotional situations or mood state.

#### 7.2.6. Inclusion of Individual and Group Analysis 

The workload and content of vigilance stimuli may vary across subjects; consequently, brain activation may differ between individuals in relation to stimuli. Researchers should consider for individual level analysis to investigate these variations. This will be beneficial in avoiding false positive and false negative outcomes in neural analysis due to the influence of systemic physiological fluctuations [246]. 

## 8. Concluding Remarks

This paper presents a comprehensive review of conventional and unconventional means of enhancement techniques on vigilance. While the findings of most conventional enhancement techniques to date were either solitary or contradictory in nature, modern unconventional enhancement techniques exerted a positive effect on vigilance, in general. Besides, the effects of enhancement were seemingly influenced by several factors, such as time-on-task and overall experimental protocol, and may not last so long. Additionally, the underlying neural mechanisms are yet to be fully elucidated. Therefore, several recommendations were made to reduce vigilance decrement and improve the experimental protocols. Understanding how and where the enhancement perceptions are generated and which cortical networks are highly affected will aid in the optimization of enhancement techniques for modulating the overall cognition and mood states. In addition, further studies with more accurate reporting of experimental protocols will also help to clarify the most promising effects. 

## Figures and Tables

**Figure 1 brainsci-09-00178-f001:**
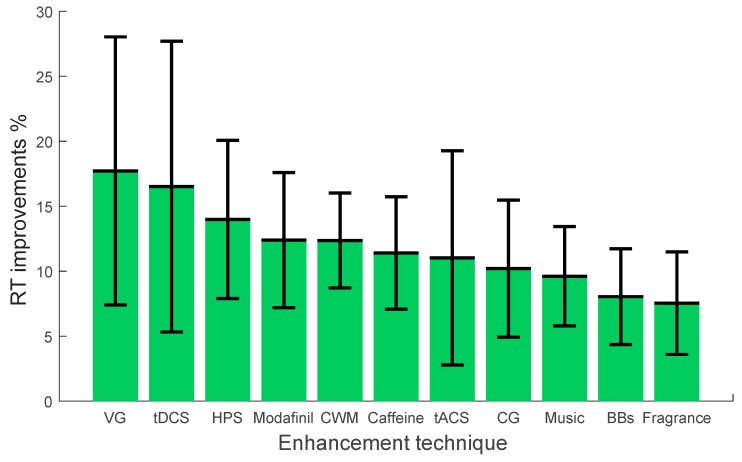
Percentage improvement on vigilance measured by RT when monotonous vigilance tasks are considered. Key to x-axis abbreviation: VG: video game playing; Transcranial direct current stimulation: tDCS; HPS: haptic stimulation; CWM: cognitive workload modulation; tACS – transcranial alternating current stimulation; CG: chewing gum; BBs: binaural beat stimulation. The error bar represents the standard deviation in the reaction time (RT) improvements across different studies.

**Figure 2 brainsci-09-00178-f002:**
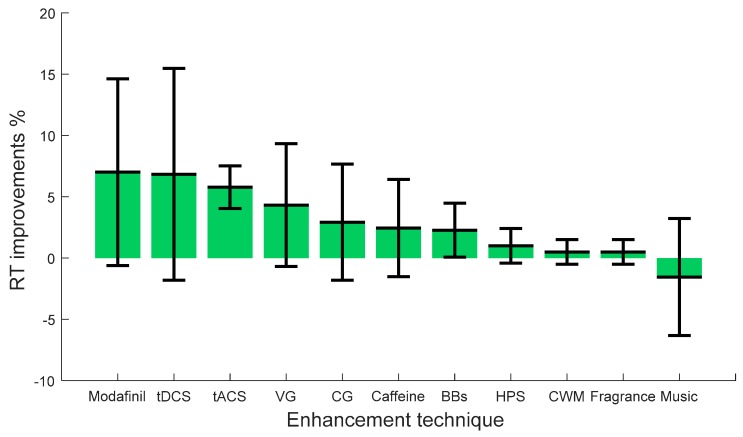
Percentage improvement on vigilance measured by RT when complex vigilance tasks are considered. Key to x-axis abbreviation: Transcranial direct current stimulation: tDCS; tACS: transcranial alternating current stimulation; VG: video game playing; CG: chewing gum; BBs: binaural beat stimulation; HPS: haptic stimulation; CWM: cognitive workload modulation. The error bar represents the standard deviation in reaction time (RT) improvements across different studies.

**Table 1 brainsci-09-00178-t001:** Studies of unconventional means of enhancement on vigilance.

Study	Enhancement Technique	Vigilance Task	No of Subjects	Summary of Results (Compare before and after Intervention)	Attributes of Effects on Vigilance	Comment
[44]	Modafinil (200, 400 mg)	Working memory and mathematical ability	50	Performance and alertness significantly improved compared to placebo.	Positive	Modafinil did not appear to offer advantages over caffeine.
[45]	Modafinil (100, 200 mg)	Memory and attention	60	Enhanced performance on tests of digit span, visual pattern, and spatial planning. Subjects reported to be more alert, attentive and energetic when on the drug.	Positive	Selectively improved neuropsychological task performance.
[46]	Modafinil (100, 200 mg)	Rapid visual information processing	89	Improved target detection, performance and reaction time.	Positive	The high IQ may limit the detection of Modafinil’s positive effects.
[47]	Modafinil (100, 200 mg)	Digit span, color naming and rapid visual information processing	60	Improved performance in all tasks and enhanced reaction time.	Positive	The benefits of Modafinil were dose-related.
[48]	Modafinil (200 mg)	Working memory, processing, and speed/attention	41	Improved performance on a test of sustained attention but no improvements on other cognitive tests.	Positive	Modafinil may be helpful in methamphetamine-dependent subjects who frequently used the drugs.
[49]	Video game (1 hour)	Visual target and random letters	30	Significant improvements on accuracy, reaction time and attention.	Positive	Video game playing enhanced the capacity of visual attention and its spatial distribution.
[50]	Video game (1 hour)	Multiple object tracking	114	Participants who played action games showed enhanced performance on all aspects of attention tested compared to non-gamers.	Positive	Suggested careful control of video game usage when assessing gender differences in attentional tasks.
[51]	Video game (40 min)	Target detection	77	Improved vigilance during training and during subsequent test phase in which no feedback was provided.	Positive	Video game-based methods may be useful for training sustained attention.
[52]	Video game (1 hour)	Multitasking, sign and drive	174	Enhanced sustained attention and working memory.	Positive	Suggest video game as a powerful tool for cognitive enhancement.
[53]	Video game (25 min)	Target detection	294	Enhanced accuracy and reduced response time and false alarms.	Positive	Active video game players (AVGPs) rate the task lower than NVGPs in terms of total or global perceived workload.
[27]	Video game (1.5 hours)	Target detection	28	Significant increase in correct detections. Improved overall performance for soldiers and students.	Positive	Video game did not induce the decrement function compared to vigilance task.
[26]	Video game (15 min)	Target detection	32	Sustained attention could be trained using the knowledge of results (KR) via video-game platform. The results indicated that KR enhanced sustained attention.	Positive	Video game environment can support effective sustained attention training in professional military and general population.
[29]	6 Hz Theta tACS (15 min)	Delayed letter discrimination task	18	Significantly improved visual memory matching and reaction times compared to placebo stimulation.	Positive	Revealed the suitability of the technique to induce coupling or decoupling of behaviour between brain regions.
[30]	40 Hz gamma tACS (12 min)	Raven’s matrices	20	Improved target detection and increased performance.	Positive	Selectively accelerated logical reasoning on the prefrontal cortex.
[28]	40 Hz Gamma tACS (30 min)	Visual target	24	Enhanced performance in vigilance tasks and significantly decreased slowdown of reaction times.	Positive	Error rate did not differ between groups.
[54]	tDCS (20 min)	go/no-go paradigm	46	No effect on performance and reaction time.	No difference	The increase of dopaminergic activity led to a deterioration of the executive function.
[32]	tDCS (10 min)	Simulated air traffic controller task	19	Behavioral measures showed a significant improvement in target detection performance compared to the sham stimulation.	Positive	The technique suggested that enhancing performance in work settings required sustained attention.
[33]	tDCS and caffeine (30 min)	Psychomotor vigilance task (PVT)	30	Enhanced vigilance and better subjective ratings for fatigue, drowsiness, energy, and composite mood.	Positive	tDCS worked better than caffeine on vigilance and mood state in which its effects lasted several hours.
[55]	tDCS (9 min)	Visual digit stimuli	23	Stimulation decreased the reaction time and increased skin conductance and arousal.	Positive	Very sensitive to arousal.
[56]	tDCS (15 min)	go/no-go paradigm	8	No effect on the accuracy or reaction time.	No difference	Could be used to enhance Theta amplitude over the frontal midline.
[57]	tDCS (10 min)	Video game/multitasking paradigm	41	Enhanced performance on multitasking paradigm and reduced its cost by 20%.	Positive	The result suggested left prefrontal cortex (PFC) in facilitating the performance of more than one task or multitasking.
[58]	tDCS (20 min)	Visuospatial task	18	Improved executive function and dual tasking in older adults with functional limitations.	Positive	Improved gait markers.
[40]	Music (1 hour)	Auditory target detection	76	Improved accuracy and detection rate.	Positive	Supported the use of music to improve vigilance in educational and clinical settings.
[59]	Music (30 min)	Attention test	89	Preferred music improved the concentration and attention level.	Positive	It is important to select music that workers strongly like to avoid negatively affecting their concentration.
[60]	Music (30 min)	Reading texts	24	Participants scored significantly lower after listening to nonpreferred music while reading.	Negative	Participants disrupted by a nonpreferred musical background.
[61]	Music (10 min)	Attention test	102	Background music with lyrics had significant negative effects on concentration and attention level.	Negative	Music with lyrics caused distractions and reduced worker attention and performance.
[62]	Music (1 hour)	Conjunction search task	12	Listening to preferred music increased performance level. Different temporal patterns were depicted in the change of performance.	Positive	Music effected emotions and mood states.
[63]	Music (21 min)	Attention to response task	158	Positive valence music significantly decreased the miss rates relative to negative valence music or silence.	Positive	Results supported the attentional restoration theory of the impact of music on sustained attention.
[39]	Music (10 min)	Attention to audio	20	Enhanced global efficiency of brain, enhanced local neural efficiency at the prefrontal lobe, and increased sustained attention.	Positive	Music directly affected cognitive system and led to improved brain efficiency.
[41]	Binaural beat (1.5, 4, 16, 24 Hz; 30 min)	Target detection	29	Beta binaural beat yielded more correct target detections and fewer false alarms than the presentation of theta/delta binaural beats.	Positive	The study’s assessment was conducted using questionnaires, which was a subjective method.
[42]	Binaural beat (18.5 Hz; 5 min)	Audio-visual light	15	Beta binaural beat enhanced in the range of 13–21 Hz and a high increase in the 18.5 Hz amplitude.	Positive	Binaural beats may potentially enhance attention.
[64]	Binaural beat (7, 16 Hz; 13 min)	Reading texts	31	No changes detected before and after binaural beat stimulation at Beta and Theta frequencies.	No difference	Short recording time of 13 minutes.
[35]	Vibrotactile (5 Hz; 3 hours)	Target detection	11	Reduced reaction time to stimuli.	Positive	No significant difference in accuracy.
[65]	Vibrotactile (250 Hz; 40 min)	Auditory or a visual display	98	The audio results showed greater performance improvement compared to the visual modality.	Positive	Visual modality posed no benefit for sustaining the performance.
[37]	Vibrotactile (250 Hz; 40 min)	Auditory or a visual display	150	Detection accuracy was significantly greater in the auditory modality compared to the visual modality. Reduced mental workload.	Positive	A rest break can restore the performance in auditory and visual vigilance tasks.
[36]	Vibrotactile (15 Hz; 16 min)	Target detection	20	Participants performed better in perceptual sensitivity and sustaining attention level.	Positive	Haptic-based brainwave entrainment poses the potential for cognitive training.

The following subsections discuss in detail the findings obtained by different studies and highlight the key differences between them with consideration of several factors that affect vigilance.

**Table 2 brainsci-09-00178-t002:** Comparison of enhancement challenges.

Enhancer Type	Response Time	After Effect Duration of Action	Translation to Real-Life Practice	Cost Per Dose	Adhering to Ethical Approval	Advantages	Dis-Advantages
Modafinil [69]	~20 min (slow)	11.5 hours (400 mg)	Difficult	Moderate	Difficult	Highly effective for higher cognitive functions. Can be used for enhancement and therapeutic purposes.	May generate possible abuse and addiction and the waking mechanism has not been fully elucidated.
Gaming [241]	~30 min (slow)	To be elucidated	Difficult	Moderate	Easy	Easily integrated into technology. Easily accessible. Real-time stimulation.	Long-time playing increases the risk of depression, aggressive behaviors, addiction and musculoskeletal pain. Requires full attention of users.
tDCS [179,242]	~10 min (fast)	6 hours	Difficult	Moderate	Easy	Portable, and tolerable User-friendly. Can easily be combined with pharmacotherapy. Can be applied to specific brain region.	Needs to be done in a quiet environment, such as the lab or clinic. May not be safe for long-term monitoring.
Music [129]	~ minutes (fast)	To be elucidated	Easy	Low	Easy	Real-time. Long-term. stimulation Used in therapy.	Affects wide cognition domain, emotions and mood state.
Binaural Beats [243]	~6 min (fast)	To be elucidated	Easy	Low	Easy	Real time. Long-term stimulation. Can enhance certain cognitive function.	Sensitive to age variations. Depends on binaural frequency.
Haptic/ tactile Stimulation [244]	~11 min (fast)	To be elucidated	Easy	Low	Easy	Real-time stimulation. Long-term stimulation. Can enhance certain cognitive functions.	Affected by skin diseases.
Challenging integration [38]	~5 min (fast)	To be elucidated	Moderate	Low	Easy	Real-time stimulation. Long-term stimulation.	Sensitive to individual difference.
Chewing Gum [211]	~minutes (fast)	20 min	Easy	Low	Easy	Real-time stimulation.	Has slow response. May cause jaw problem.
Caffeine [25,44]	~ minutes (fast)	2 hours (200 mg)	Easy	Low	Easy	Widely accepted by society. Has faster response. Doses intake can be easily controlled.	Reported symptoms of nervousness, excitation, pain, dry mouth, tremor, nausea, and jitteriness.
Fragrance/ Odour [245]	~ minutes (fast)	To be elucidated	Easy	Low	Easy	Real-time stimulation. Has faster response.	May affect skin. Causes headache. Have poisonous effects on the brain and nervous system.

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
