# Peer review of "Vigilance Decrement and Enhancement Techniques: A Review"

_brainsci, 2019, doi:10.3390/brainsci9080178_

Round 1
Reviewer 1 Report
The additions to the paper since the initial submission have strengthened it. The comprehensive summary is a great idea, but I have comments on execution. Systematically distinguishing between complex and monotonous vigilance tasks is a good approach. The bar plots are a useful addition. Please add a key to the x-axis abbreviations in the caption for each figure. Please also include a definition of your error bars in the caption.
A do have a critique of the comprehensive summary, which relates to the point above about the error bars. It isn't clear that you are discussing the variance over studies appropriately. For instance, in the complex vigilance bar plot, most of the error bars include zero. This seems to indicate that, on the whole, very few interventions improve vigilance on these tasks. But this is not how you write about it. In general, these sort of summary discussions need to handled rigorously. You are effectively reporting a meta analysis and there are techniques and best practices that should be followed.
In addition, on lines 549--550, you write: "The improvements of action video games (VG) on sustained attention is due to their potential in altering the neural connections in the brain."
It is not clear if you are putting this forth as a statement of well-established fact or as a hypothesis. If it is a hypothesis, better to write: "The large improvements to vigilance associated with action video games are consistent with prior work [citations] that associates video game play with neuroplasticity in ...".
Reading the manuscript again for this review I noticed more issues with English language that made the paper less clear. There are too many small instances to note, but I did find myself itching to mark up and revise as I read.
In summary, my major concern is that the information reported in the comprehensive summary has not been handled with enough rigor to make inferences from the quantitative data that is being summarized. I feel that this section should be scaled up to be statistically rigorous, or scaled back so that it remains descriptive. By adding error bars to the figures without description of the variance or statistics used, it invites inappropriate inference. If the error bars are literally the range of the data you aggregated, simply stating this and describing what you are and are not doing in this presentation of aggregated data at the beginning of the section, and what inferences should and should not be drawn, will go a long way towards the correct interpretation of your work.
Author Response
Point 1: “The additions to the paper since the initial submission have strengthened it. The comprehensive summary is a great idea, but I have comments on execution. Systematically distinguishing between complex and monotonous vigilance tasks is a good approach. The bar plots are a useful addition. Please add a key to the x-axis abbreviations in the caption for each figure. Please also include a definition of your error bars in the caption.”
Response 1: We have added the abbreviations to the captions and define the error bars in the caption with the standard deviation error. (Please see the attachment)
Figure 1. Percentage improvement on vigilance measured by RT when monotonous vigilant tasks are considered.
Key to x-axis abbreviation: VG - video game playing; Transcranial direct current stimulation – tDCS; HPS - haptic
stimulation; CWM - cognitive workload modulation; tACS – transcranial alternating current stimulation; CG -
chewing gum; BBs - binaural beat stimulation. The error bar represents the standard deviation in the reaction time
(RT) improvements across different studies.
Figure 2. Percentage improvement on vigilance measured by RT when complex vigilant tasks are considered. Key
to x-axis abbreviation: Transcranial direct current stimulation – tDCS; tACS – transcranial alternating current
stimulation; VG - video game playing; CG - chewing gum; BBs - binaural beat stimulation; HPS - haptic
stimulation; CWM - cognitive workload modulation; . The error bar represents the standard deviation in reaction
time (RT) improvements across different studies.
Point 2: “A do have a critique of the comprehensive summary, which relates to the point above about the error bars. It isn't clear that you are discussing the variance over studies appropriately. For instance, in the complex vigilance bar plot, most of the error bars include zero. This seems to indicate that, on the whole, very few interventions improve vigilance on these tasks. But this is not how you write about it. In general, these sort of summary discussions need to handled rigorously. You are effectively reporting a meta analysis and there are techniques and best practices that should be followed.”
Response 2: We have updated the comprehensive summary to address this comment. We would like also to highlight that only few studies have used HPS, CWM and Fragrance stimulation to enhance vigilance with complex tasks. Most of these stimulation methods were used in simple vigilance tasks. Below, please find the updated comprehensive summary section:
5. Comprehensive summary
After reviewing the nature of the stimulating tasks and the contexts in which these studies were conducted, we now present a critical integrating summary on the effectiveness of enhancement techniques on vigilance. This section focuses mainly on highlighting the attributes of effects on vigilance (positive, negative or no significant impact on vigilance). Highlighting the effects of various enhancement techniques on different types of vigilance tasks may help researchers/scientists to select the appropriate enhancement technique in the context of their study and may help in reducing the contradictory findings. The comprehensive summary is derived from all the studies discussed in the previous sections including the onse listed in Table 1. We categorize the vigilance tasks into two types: monotonous and complex. We define the monotonous tasks as the ones that require to sustain attention such as visual monitoring and target detection. On the other hand, we define complex vigilance tasks as the ones that are more cognitively demanding, e.g. comprehension and working memory. We have conducted this review in the context of job/work, monitoring/ surveillance, driving, learning, cognition and memory tasks, typing, shopping, and sports. The percentage improvements in vigilance measured by reaction time (RT) for each enhancement technique are depicted in Figure (1) and Figure (2) for monotonous and complex vigilance tasks respectively. Positive RT improvements indicate the enhancement technique reduced the reaction time to stimuli while negative RT improvements increased or impaired the reaction time. Zero RT improvements indicates that the enhancement technique did not have any significant effect on vigilance. For each enhancement method, the error bar represents the standard deviation from the percentage RT improvement mean of the extracted data from various studies.
To the best of our knowledge, when simple monotonous vigilance tasks were considered, all enhancement techniques showed a positive impact on vigilance. The techniques showed average improvement varying between +8% and +18% as measured by the percentage reaction time (RT) improvement depicted in Figure 1. The highest improvements (more than +15%) on monotonous vigilance tasks were reported when video game and transcranial direct current stimulations were used [32,33,49,52]. Haptic stimulation (HPS), modafinil, cognitive work-modulation (CWM), tACS, chewing gum (CG) and caffeine [28,29,35,38,46,164,206] showed an improvements of more than +%10. However, music, binaural beats (BBs) and fragrance produced improvements of equal or less than +10% in the simple monotonous vigilant tasks [41,123-125,193].
The large improvements in vigilance associated with action video games are consistent with prior work [215] that associate video game play with neuroplasticity in the afferent brain regions and neural networks. This involves specifically the areas associated with empathy,
motor executive function, decision-making, and emotion regulation. Such modulation may thus enhances the communication between brain regions responsible for attention and results in improving performance. In addition, engaging action video games increase the arousal level of players by the adrenergic route, which in turns make the motor behaviour more responsive decreasing the RT to the incoming stimuli and increasing attentional abilities [216]. In particular, research has suggested that playing AVGs can lead to improvements in perceptual [217], visuo-spatial [218], perceptuo-motor [219] and attentional abilities [220], and that such improvements can also extend to cognitive control functions such as cognitive flexibility [221], and working memory updating [222], but not inhibitory control [223]. Similarly, the improvements in vigilance using tDCS may be due to altering the membrane potential and spontaneous firing rates [224]. This in turn increases the connectivity in the prefrontal cortex, dorsolateral and other remote brain regions. The modulation of targeted brain cortices led to improved performance. Likewise, the positive impact of HPS on vigilance is due to the changes induced on somatosensory cortex such as representational map reorganization [150]. HPS studies have demonstrated that the stimulation of tactile afferent fibers and essential neuroanatomical elements of affective touch activates specific brain areas. This activation pattern is influenced by subject’s attention. The changes in the functional brain network enhance the information transfer and attention.
Additionally, the improvements on vigilance with modafinil is consistent with studies on adaptive inhibition, see review [225]. Modafinil exerts its action by competitively binding to the dopamine transporter as well as by inhibiting norepinephrine uptake. This produces an overall elevation of catecholamine levels and potentiation of adrenergic neurotransmission. Stimulants enhance attention by increasing neuronal activation or by releasing neuromodulators, facilitating the synaptic changes that underlie learning/attention. Overall, the increment in the neural activations and connectivity patterns improved performance on simple and complex vigilance tasks.
Cognitive work-modulation increases the workload to a certain level, which leads to increasing the engagement of attentional resource and enhance performance on simple vigilance tasks. In the same way, the positive impact of tACS on vigilance could be explained by observing that tACS targets the brain's natural electrical oscillations, which represent neuronal patterns of communication throughout regions of the brain [226]. This is unlike tDCS, which targets brain structures, such as particular regions of the cortex. Therefore, it permits physiological entrainment through frequency stimulation at nearly imperceptible current strengths. Consequently, the improvements of BBs on vigilance may be due to BBs producing phase synchronization across the cortex [147]. The increase in phase synchronization in the auditory cortex facilitates the neural communication, promotes neural plasticity, and enhances the overall performance.
The positive impact of CG on vigilance could be explained as due to the fact that it increases the heart rate, which leads to increasing the flow of nutrients, such as glucose, to the brain [227]. The positive effect of CG on vigilance tasks may also be due to either: (1) CG restoring arousal after a vigilance task reduces arousal to a sub-optimal level, or: (2) CG reducing arousal after a vigilance task heightens arousal to an excessive level [228]. Correspondingly, the improvements of music on simple vigilance tasks is in line with a study that used music to regulate people’s mood and arousal level [229]. Literature has reported listening to preferred music reduces stress during driving and lowers emotional arousal under frustrating circumstances, such as heavy road congestion [126,143]. In this context, comfortable volume of background music exposure improves one’s performance while performing simple vigilance tasks. One may also note that music has a positive impact on emotional reactions and achievement in sports. Moreover, the positive impact of fragrance on vigilance could also be explained due to its effectiveness in alerting mood and feelings [183].
Relaxing fragrance affects performance efficiency and feelings, since subjects who are tense and uncomfortable may find it hard to concentrate on the task.
On the other hand, when complex tasks are considered, smaller improvements with contradictory findings on average of -2% to +7% are found as shown in Figure 2. For this type of vigilance enhancement tasks, Modafinil, tDCS, tACS and VG showed the highest improvements to vigilance compared to the other enhancement techniques. Interestingly, tACS has shown consistent positive impact on the two types of vigilance tasks (simple and complex), with an average improvement of more than +6% on complex vigilance tasks. Similarly, BBs, HPS, CWM, and fragrance showed both positive impact and no significant impact on vigilance. Unfortunately, the number of studies for the majority of these contexts and tasks was small, rendering the respective conclusions somewhat unreliable. Likewise, CG and caffeine reported positive and negative impacts on vigilance with average improvements of more than +3% on complex vigilance tasks. Meanwhile, some of the complex tasks showed impaired performance with caffeine and CG [230,231]. This may reflect the effects of increased alertness or chewing interfering with sub-vocal rehearsal.
Modafinil appears to enhance sustained attention, learning and memory. Negative cognitive consequences of modafinil intake were also reported in a small minority of complex tasks, though not consistently on anyone [225]. Although there is an impairment on the reaction time to stimuli in some tasks, low doses of modafinil have been shown to enhance working memory in healthy test subjects, especially at moderate task difficulties and for lower-performing subjects. Even though the mode of action of modafinil is not yet understood, it may be noted that modafinil enhances adaptive response inhibition, making the subjects evaluate a problem more thoroughly before responding, thereby improving the overall performance, reaction time and accuracy.
Negative consequences of VG were also reported in a small minority of complex vigilance tasks especially with children at school [232,233]. Playing VG showed decreased students’ performance at school, which might be due to developing attention problems and poorer sleep quality. In this context, attention deficits and poor sleep could both plausibly impair academic performance. Moreover, music has been found to negatively impact complex vigilance tasks. For example, in comprehension it reduced performance down to -7%, regardless of the type and condition (fast tempo, classical, and familiarity) of the music [60,61]. Positive effects (better results with music) were found only for simple math tests [234]. Background music with complex vigilance tasks acts as a distraction when it comes to human vigilance performance. Even though music has been shown to benefit driving performance and behavior, it may still be a major distraction and detrimental to driving abilities, e.g. in loud volumes. It is suggested that loud volume might influence vigilance due to its greater processing demands on the central nervous system. Attention may be deterred from the primary task/task at hand, thus causing an impaired RT. An alternative reason is that such high sound volume may cause an anxiety effect within subjects. It is well documented that chronic exposure to noise increases stress levels. Another possible moderator is the habituation. Since music media are increasingly available everywhere, its effects may diminish over time. The review highlights that the negative impact of music on complex vigilance tasks (such as comprehension) is a general effect.
Looking at the nature of various vigilance tasks, it is quite safe to make a conclusion regarding the effectiveness of enhancement techniques. Consistent positive effects were found only with simple vigilance tasks. VG, tDCS, tACS, and modafinil work well in enhancing vigilance regardless of the type of tasks compared to other enhancement techniques. However, there is variability across subjects in the extent to which stimulation modulates behavior. This provide a challenge for the development of applications. For example, a large part of variability in the after-effects of motor cortical tDCS may be due to the inter-individual differences in the
electric fields and size of electrodes. In this context, we anticipate that individualized electric field dosimetry could be used to control the neuroplasticity effects of tDCS. It is suggested that an effort should be made to develop more specific theories about the impact of cognitive enhancement and to increase the methodological quality of relevant studies. In fact, individual differences in response to the aforementioned vigilance enhancement techniques are also reported. These individual differences are often attributed to differences in subjects’ routine use, number of doses, culture and preference. There may be other individual differences, for example personality and age that are known to influence performance. Researchers have thoroughly investigated other variables, including temporary individual and stable variations. Temporary differences comprise coping strategies and fatigue, while stable differences include sleep behavior, gender and introversion [235-238]. Various research groups have proposed alternative neurophysiological and biological biomarkers to investigate vigilance tasks. They include electrical neural activity, salivary melatonin levels, circadian rhythm, heart rate variability, and cerebral blood flow. Some of these neurobiological implications can be influenced by training, such as deep breathing and biofeedback exercises. Hence, it is possible that given such training, individuals may be able to handle more prolonged or more complex vigilance tasks, however, few such studies have been performed.
Point 3: “In addition, on lines 549--550, you write: "The improvements of action video games (VG) on sustained attention is due to their potential in altering the neural connections in the brain."It is not clear if you are putting this forth as a statement of well-established fact or as a hypothesis. If it is a hypothesis, better to write: "The large improvements to vigilance associated with action video games are consistent with prior work [citations] that associates video game play with neuroplasticity in ...".”
Response 3: We have updated the sentence into the following: “The large improvements to vigilance associated with action video games are consistent with prior work [209] that associates video game play with neuroplasticity in the afferent brain regions and neural networks.”
Point 4: “Reading the manuscript again for this review I noticed more issues with English language that made the paper less clear. There are too many small instances to note, but I did find myself itching to mark up and revise as I read.”
Response 4: The paper has been thoroughly edited and a number of words/sentences have been substituted or re-phrased appropriately, as per the reviewer’s suggestion.
Point 5: “In summary, my major concern is that the information reported in the comprehensive summary has not been handled with enough rigor to make inferences from the quantitative data that is being summarized. I feel that this section should be scaled up to be statistically rigorous, or scaled back so that it remains descriptive. By adding error bars to the figures without description of the variance or statistics used, it invites inappropriate inference. If the error bars are literally the range of the data you aggregated, simply stating this and describing what you are and are not doing in this presentation of aggregated data at the beginning of the section, and what inferences should and should not be drawn, will go a long way towards the correct interpretation of your work.”
Response 5: We have updated the entire section of the comprehensive summary as reported in Response 2. We would also like to highlight that, error bars in Figure 1 and Figure 2 represent the standard deviation of the data extracted from the various studies (in this case; the percentage of change in the reaction time to stimuli). In this section, we are mainly interested in highlighting the attributes of the effects on vigilance (positive, negative or no significant impact).

Reviewer 2 Report
Thank you for preparing a revised version of this manuscript. It is evident that many of the comments have been addressed, however, there are also several that have not been addressed. As there is no letter of response to reviewers I cannot tell whether this is an omission or whether there is a reasoned justification.
The inclusion of the new comprehensive summary is positive, but this section does not contain any citations, so it is not clear whether the assertions made are speculation or somehow grounded in previous work or theory. I feel this is a major omission and weakness in a scientific piece of writing (it is not opinion, but interpretation of findings).
The manuscript still contains errors related to grammar, spelling, and the use of citations.
Some of the edits seem to be hasty and address the comments in the review only superficially. There are several statements that make reference to something raised in the review (e.g. individual differences) but they don't actually discuss or address these in any depth. I feel this is not a comprehensive way to improve a manuscript.
Author Response
Point 1: “Thank you for preparing a revised version of this manuscript. It is evident that many of the comments have been addressed, however, there are also several that have not been addressed. As there is no letter of response to reviewers I cannot tell whether this is an omission or whether there is a reasoned justification.”
Response 1: We thank the reviewer for pointing this out. In this response letter to the reviewers’ comments, we tried to address all the comments and suggestions.
Point 2: “The inclusion of the new comprehensive summary is positive, but this section does not contain any citations, so it is not clear whether the assertions made are speculation or somehow grounded in previous work or theory. I feel this is a major omission and weakness in a scientific piece of writing (it is not opinion, but interpretation of findings).”
Response 2: The comprehensive summary is based on the studies discussed in the previous sections of the manuscript. We have included the citations as suggested by the reviewer.
Point 3: “The manuscript still contains errors related to grammar, spelling, and the use of citations.”
Response 3: The paper has been thoroughly edited and a number of words/sentences have been replaced or re-phrased appropriately, as per the reviewer’s suggestion. We have also carefully reviewed the references, and now confirm their consistency across the entire manuscript.
Point 4: “Some of the edits seem to be hasty and address the comments in the review only superficially. There are several statements that make reference to something raised in the review (e.g. individual differences) but they don't actually discuss or address these in any depth. I feel this is not a comprehensive way to improve a manuscript.”
Response 4: We have added few paragraphs at the end of each sub-section as follow:
3.3. Transcranial direct current stimulation
In addition, stimulation parameters such as duration, intensity, frequency, electrode-position, and control settings can also modulate the outcome of the tDCS effect. More
importantly, inter- and intra-individual differences, including genetics, age, gender, physiological differences and baseline task performances, all imply the importance of a certain neural state. This state amy determine the modulating effect on stimulated individuals through its interaction with tDCS [115]. It is suggested that the baseline state of each individual is different and that one’s receptivity to the tDCS changes with his/her baseline performance. It is also important for studies to choose an applicable baseline on which to evaluate the effect of tDCS. In addition, the task difficulty is another contributor to the state-dependent nature of the effects of tDCS. In a cognitive control task, the impact of tDCS was observed in the easy and medium difficulty conditions, but not in the case of the most difficult ones [57]. Thus, this review implies that tDCS effect is interactive and state-dependent. The task difficulty and consistent individual differences modulate one’s responsiveness to tDCS, while researchers’ choices of independent behavioral baseline measures can also critically affect how the effect of tDCS is evaluated. These factors are likely the key contributors to the wide range of variability in tDCS effects between individuals, stimulation protocols, and between different studies in the literature. Future studies using tDCS, and possibly tACS, should take such state-dependent condition in tDCS responsiveness into account.
3.4. Music
One potential approach used for explaining the impact of background music on reading performance is the effect of lateralization. It is assumed that an increase in the activation of one’s brain hemisphere decreases the activation of the other region [140,141]. If background music activates the right hemisphere, the performance in tasks that need a highly activated left hemisphere, such as verbal tasks, could deteriorate. In this context, listening to music is considered as dual-task processing. Thus, it not only involve listening but also the verbal abilities and interferes with reaction time. Another candidate explanation for the negative impact of background music on memory processes might start with deliberations on the role of attentional limitations: listening to music while performing some cognitive task might distract attention from that task and therefore impairs performance, especially in tasks that require conscious efforts [142].
Previous studies have already observed that individual differences in personality and temperamental dimensions may play an important role in music preferences, exposure to different genres, music listening habits and use. However, even considering the strong emotional impact of music on humans, these affective responses are highly specific to cultural and personal preferences. Large individual differences are observed across individuals and on how music is experienced.
Researchers must take into account individual differences when investigating the effects of music on employees’ fatigue and work tasks. The individual differences were essential in a pre-test/post-test control group study carried out on 33 air traffic controllers [143]. The subjects completed trait anxiety and extroversion measurements, as well as a diagnostic stress inventory before the formal study. Results showed a significant reduction in stress level, when the subjects listened to music. Nevertheless, an interaction effect was exhibited for individuals with high trait anxiety and introversion, who did not demonstrate a reduction in anxiety. One of the possibilities is that the more an individual listens to music, the more likely they will experience an increasingly strong emotional response to music. In other words, the more knowledge one has of music, the more the emotional responsiveness. Without accounting for individual differences, the researchers may be missing an important work environment-person factor. Therefore, future studies regarding effect of music on vigilance should take into consideration the impact of valence, tempo, familiarity, and personal preferences during the design and analysis.
4.1. Caffeine
Individual differences in response to caffeine are also commonly reported [178-180]. Researchers often point to substantial individual differences in behavioral responses to explain inconsistencies among experimental findings. These individual differences are often attributed to differences in subjects’ regular use and the number of doses. It has been thought that consumers of high daily amounts of caffeine might have acquired tolerance to behavioral effects of caffeine and are therefore less sensitive to its effect on behavior. However, little is known about the relationship between subjects’ regular caffeine intake and their behavior sensitivity to an acute dose of caffeine. There is some evidence that daily administration of 300 mg of caffeine can produce tolerance to its subjective effects in humans [181].
There is no evidence to date that demonstrates that normal consumption levels of caffeine relate to acute behavioral responses in human. A study in [182] examined subjects’ reports of typical daily consumption in the study of individual differences in response to caffeine. The study concluded that subjects’ caffeine use could not account for individual differences, but subjects’ expectancies about the effect of caffeine did predict individual differences in response to the drug.
4.2. Fragrance admisteration
One possible reason for the lack of consistency in previous findings is that existing individual differences in olfactory sensitivity have been overlooked. In particular, sensory sensitivity plays a major role in shaping the response of such individuals. A few studies have explored individual differences in olfactory sensitivity and found approximately 20 per cent of the population self-reported as having a heightened sensitivity to scent, in comparison to 70 per cent who self-categorized their sense of smell as normal [198,199]. Recent findings revealed an automatic suppression mechanism for individuals sensitive to smell [200]. Combined, these findings suggest that a significant proportion of the population is sensitive to smell. Another study [201] used the impact of odor (AIO) scale to measure the importance of smell on an individual’s liking of people, places and products. It was found that high AIO scores were associated with odor-related memory, and attention to odor. This also influenced their liking or disliking of people based on their odors. Similar study in [202] investigated these effects across individuals based on their olfactory sensitivity. It found that odors play a significantly stronger role in individuals sensitive to smell on perceptions of place and people in a service setting, as well as on cognitive processes such as attention, memory and emotions.
This manuscript is a resubmission of an earlier submission. The following is a list of the peer review reports and author responses from that submission.
Round 1
Reviewer 1 Report
This is an interesting topic, and a review of relevant literature would be a very useful contribution to the literature.
I feel there are some strengths but also several weaknesses in this piece of work. Overall the review of literature seems to be comprehensive, and the organisation of information is methodical.
Despite the interesting topic and its potential value I have several concerns about this review and do not think it is currently suitable for publication.
The information is largely descriptive of study aims and findings, but does not offer much in the way of critically integrating those findings to provide the reader with a summary and a differentiated understanding. As an example, in the overview of how music is used for enhancing vigilance there is no discussion about whether the effectiveness of music depends on the nature of the task that is being performed, whether there is potential for verbal interference, and the potential for emotional responses (though these are mentioned later). It seems that because a wide selection of research is being described, where there are disparate findings these are not well interpreted and integrated due to the vastly different research aims and protocols. It is important to be clear about differences between vigilance in otherwise monotonous tasks such as visual monitoring, and vigilance in tasks that are more cognitively demanding (as is the case for cognitive enhancement potential of drugs used by students to improve academic performance). For each section/type of enhancement it would be useful to have some concluding comments that give a clear overview about what types of tasks the enhancement might benefit and why. I feel there is insufficient detail and discussion of findings to draw meaningful conclusions (other than that results vary).
The review would benefit from a discussion that addresses the role of distraction as an effective enhancer of vigilance. In this context it is also important to consider the role or re-allocation of attention to the primary task after the “enhancing” distraction, and whether the effect is temporary, whether there is potential for habituation etc. This is particularly relevant in the description of haptic feedback as a form of vigilance enhancer, for example in drivers. It is also necessary to address the potential for repeated stimulation of this nature to become stressful (and therefore unhelpful). The issue of habituation is mentioned in relation to fragrance administration, but should be discussed in more depth.
There is no mention about individual differences in response to different vigilance enhancement methods, which is, to my mind, an important omission as the aim of the paper seems to be to provide advice about what techniques are useful. In the section on safety there is mention of subjectivity, but this is described superficially.
There are many vague statements, especially in the sections about challenges. The risks and benefits mentioned in line 523 are not useful without specifying the context. Similarly it is not clear what is meant by individuals having the right to choose their own preferences. For the most part people do choose whether or not to drink caffeinated drinks or chew gum, listen to music etc. This needs better explaining. It is also unclear what the relevance of privacy and data protection is in this context. Another example is in the section about cost. There is mention of cognitive enhancement (the main focus of the paper is vigilance, not cognitive enhancement) and how this may reduce the cost of years of education. This point is very confusing as it suggests that using cognitive enhancers could substitute further or higher education. I am sure this was not the intention as it is clear that the purpose and effect of a cognitive enhancer is entirely separate to the purpose of educational training.
The comparison of the enhancement challenges in Table 2 includes some gross over-simplifications. As an example the category “drugs” presumably includes more than wakefulness drugs such as modafinil, yet the disadvantages listed seem to be specific to that drug.
In the recommendation section the reference to “task at hand” (Line 610) is unclear. What is the task? Many different task scenarios were covered in the reviewed literature.
The terminology can be confusing. For example the reference to “conventional” enhancement is unclear. Conventional in what sense? (e.g. line 378). Is Fragrance administration conventional in the same way that caffeine is?
The writing would benefit from rigorous proof reading as there are errors in grammar and use of some vocabulary. This makes the writing difficult to follow at times. Examples include not using pronouns and articles appropriately, a consistency of tense, and repeated use of “besides” to begin sentence, referring to research as “works”, not using plurals correctly.
The citations are inconsistent as sometimes the author names are used, but mostly only the reference number is provided. For ease of reading it tends to be better to use the author names when these are being referred to in the sentence, rather than just the number. One example of numerous instances is line 212 “Similarly, the study by [110] has applied…” The author name(s) followed by the reference number would make this easier to read.
Line 409 – what is the relevance of haemodyalisis? Some more context would make this information more useful.
All footnote numbers should be in superscript.
Reviewer 2 Report
A very comprehensive and well organized review of the literature on vigilance. Pros and cons of various strategies are presented, and the extent to which more research is needed on certain topics is clearly outlined. Recommendations for application of vigilance enhancing strategies and for future research are thoughtful and a good way to conclude the review.
There are minor typos---"gaming" is misspelled in Table 2, and there are a few sentence that a native English reader would catch as slightly ungrammatical. Nothing that makes the paper a challenge to read, but if fixed it would elevate the paper.
The Tables are comprehensive and helpful. However, as formatted, there are places where words are being wrapped at inappropriate places. These Tables will be difficult to format "perfectly", but putting in the effort here will make a bit difference.
On line 148 there is an in-text citation that is formatted unlike others in the paper.